# Revisiting the applicability and constraints of molybdenum and uranium-based paleo redox proxies: comparing two contrasting sill fjords

5    K. Mareike Paul[1], Martijn Hermans[1,2], Sami A. Jokinen[3], Inda Brinkmann[4,5], Helena L. Filipsson[4], Tom Jilbert[1]

[1]Environmental Geochemistry Group, Department of Geography and Geosciences, Faculty of Science, University of Helsinki, Helsinki, 00560, Finland
[2]Baltic Sea Centre, Stockholm University, Stockholm, 114 18, Sweden
[3]Marine Geology, Geological Survey of Finland (GTK), Espoo, 02151, Finland
10    [4]Department of Geology, Faculty of Science, Lund University, Lund, 223 62, Sweden
[5]Department of Glaciology and Climate, Geological Survey of Denmark and Greenland, Denmark

*Correspondence to:* K. Mareike Paul (mareike.paul@helsinki.fi)

**Keywords:** deoxygenation, enrichment factors, sequential extraction, sedimentary trace metals

**Abstract**

Sedimentary molybdenum (Mo) and uranium (U) enrichments are often used as redox proxies to reconstruct bottom water redox changes. However, these redox proxies may not be equally reliable across a range of coastal settings due to varying depositional environments. Fjords vary greatly in their depositional conditions, due to their unique bathymetry and hydrography, and are highly vulnerable to anthropogenic and climatic pressures. Currently, it is unknown to what extent Mo and U sequestration is affected by variable depositional conditions in fjords. Here, we use pore water and sequential extraction data to investigate Mo and U enrichment pathways in sediments of two sill fjords on the Swedish west coast with contrasting depositional environments and bottom water redox conditions. Our data suggest that sedimentary authigenic Mo and U pools differ between the two fjords. At the ir/regularly dysoxic (oxygen = 0.2–2 mL $L^{-1}$) Gullmar Fjord, authigenic Mo largely binds to manganese (Mn) oxides and to a lesser extent to iron (Fe) oxides; Mo sulfides do not play a major role due to low sulfate reduction rates, which limits the rate of Mo burial. Authigenic U largely resides in carbonates. At the ir/regularly euxinic (oxygen = 0 mL $L^{-1}$; total hydrogen sulfide $\geq$ 0 mL $L^{-1}$) Koljö Fjord, authigenic Mo is significantly higher due to binding with more refractory organic matter complexes, and Mo-Fe-sulfide (S) phases. Uranium is moderately enriched and largely bound to organic matter. We found no direct evidence for temporal changes in bottom water redox conditions reflected in Mo and U enrichments at either Gullmar or Koljö Fjord. While sulfidic bottom waters favor Mo sequestration at Koljö Fjord, enrichment maxima reflect a combination of depositional conditions rather than short-term low oxygen events. Our data demonstrate that secondary pre- and post-depositional factors control Mo and U sequestration in fjords to such an extent that bottom water redox conditions are either not being systematically recorded or overprinted. This explains the large variability in trace metal enrichments observed in fjords and has implications for applying Mo and U as proxies for environmental redox reconstructions in such systems.

**1 Introduction**

Sedimentary molybdenum (Mo) and uranium (U) enrichments are frequently used as (paleo) redox proxies to reconstruct changes in bottom water oxygen ($O_2$) due to their redox-sensitive geochemical behavior (Algeo and Lyons, 2006; Jokinen et al., 2020b; Bennett and Canfield, 2020). However, the reliability of these redox proxies may be biased by inadequate understanding of Mo and U enrichment pathways and secondary depositional environmental factors (Bennett and Canfield, 2020; Jokinen et al., 2020b; Paul et al., 2023). Besides bottom water redox conditions, secondary factors such as "the basin reservoir effect" and equilibrium with $FeMoS_4$ (Algeo and Lyons, 2006; Helz, 2021), particulate iron (Fe) and manganese (Mn) (oxy)(hydr)oxide "shuttling" (Fe and Mn oxide shuttling hereafter;

Crusius et al., 1996; Algeo and Tribovillard, 2009), reoxygenation events (Zheng et al., 2002a,b, Morford et al., 2009), the depth and intensity of the sulfate–methane transition zone (SMTZ) in the sediment (Jokinen et al., 2020b), sedimentation rate (Algeo and Maynard, 2004; Liu and Algeo, 2020), and local detrital background (Van der Weijden, 2002; Brumsack, 2006) may considerably control authigenic Mo and U sequestration in modern coastal sediments (Jokinen et al., 2020b; Bennett and Canfield, 2020; Paul et al., 2023).

Bottom water deoxygenation is expanding in coastal areas globally due to rising anthropogenic and climatic pressures (Breitburg et al., 2018; Conley et al., 2011; Meier et al., 2022). The severity of deoxygenation varies in response to individual properties of coastal depositional environments, such as water mass restriction, temperature and salinity induced density gradients, productivity, and sedimentation rate. Standard thresholds to designate the degree of deoxygenation are dysoxic ($O_2$ = 0.2–2 mL $L^{-1}$), suboxic ($O_2$ = 0 mL $L^{-1}$), and euxinic ($O_2$ = 0 mL $L^{-1}$; total hydrogen sulfide, $\sum H_2S$ = > 0 mL $L^{-1}$) after Algeo and Li (2020) and references therein.

Fjords are particularly sensitive to anthropogenically-, and climatically induced environmental changes, due to their unique morphological, hydrological, and sedimentological characteristics (Howe et al., 2010; Bianchi et al., 2020 and references therein). Fjords are formed by glacial erosion from Late Cenozoic ice sheets in mid–high latitudes. Typically, fjords are long, narrow, deep, and steep-sided U-shaped estuaries that often have one or more sill(s) (Pickard and Stanton, 1980; Syvitski and Shaw, 1995). Such sill fjords often experience a strongly limited water mass exchange between the deep basin(s) and the coastal ocean, resulting in episodic to permanent bottom water deoxygenation. Deoxygenation in fjords is further aggravated by anthropogenic climate change (e.g., ocean warming), high riverine and coastal runoff of nutrients and organic matter (OM), which all lead to strong vertical water mass stratification, increasing eutrophication and primary productivity, and a higher $O_2$ demand than $O_2$ supply to the bottom water upon aerobic degradation of OM (Aksnes et al., 2019; Boone et al., 2018; Darelius, 2020). Yet, high sediment accumulation and organic carbon ($C_{org}$) burial rates (e.g., Bianchi et al., 2020; Smith et al., 2015) make fjords effective spatial and high temporal resolution sedimentary archives of past environmental changes (e.g., Nordberg et al., 2001; Harland et al., 2004; Howe et al., 2010; Asteman et al., 2018). Thereby, fjords are ideal to test and apply trace metal proxies for investigating deoxygenation (Russell and Morford, 2001; Goldberg et al., 2012; Brinkmann et al., 2023b).

Recent research suggests that Mo and U enrichment factors (EFs) in fjord sediments show a large range (Fig. 1c and d) and less accurately record bottom water redox changes compared to less dynamic restricted basin sediments, for which Mo- and U-EFs have mostly been applied (Paul et al., 2023). That study showed that such limitations can be partially explained by particulate Fe and Mn oxide shuttling and pore water chemistry. However, neither the mechanisms nor extent to which Fe and Mn oxide shuttling and pore water chemistry control Mo and U sequestration in

fjord sediments are fully understood. This has implications for interpreting e.g., trace metal redox proxy data derived from fjords compared to other non-fjord depositional environments.

Here, we investigate Mo and U sequestration pathways in two sill fjords with contrasting bottom water redox conditions and depositional environments using pore water and sequential extraction data of Mo, U, Mn, Fe, calcium (Ca), aluminum (Al), and sulfide (S), complemented with historical data of $O_2$ conditions and their relationship to climatic indices. We assess the applicability and constraints of Mo- and U-based redox proxies to reconstruct deoxygenation in fjord-type systems. Using these data, we aim to explain the observed wide ranges in Mo and U

enrichments in fjord settings (Paul et al., 2023). Our study demonstrates that improving the understanding of sedimentary Mo and U redox dynamics in different coastal settings is essential for a more reliable application of Mo- and U-based redox proxies for environmental reconstructions in fjord-type systems.

## 2 Materials and methods

### 2.1 Study area and bathymetrical characteristics

Gullmar- and Koljö Fjord are two adjacent sill fjords on the Swedish west coast (Fig. 1a). Gullmar Fjord (Swedish: Gullmarsfjorden) is 29 km long, 1–3 km wide and has a maximum depth of 120 m (Alsbäck Deep; Lindahl and Hernroth, 1988). Koljö Fjord belongs to an open-ended fjord system surrounding the Orust and Tjörn islands (Björk et al., 2000; McQuoid and Nordberg, 2003; Fig. 1a). Compared to Gullmar Fjord, Koljö Fjord is shallower (maximum depth of 56 m) and more restricted: it has three shallow sills to the adjacent Havsten Fjord (S1) at 12 m water depth,

Skagerrak (S2) at 8 m water depth (Nordberg et al., 2001), and Gullmar Fjord (S3) at < 5 m depth (Filipsson and Nordberg, 2004a).

### 2.2 Oxygenation history and deoxygenation drivers

Over the past century, deoxygenation has been frequently recorded in both fjords. However, the severity and duration of deoxygenation differs between the two fjords, as evident from trace metal proxy data (Paul et al., 2023). These show

lower sedimentary Mo and U enrichments Gullmar Fjord (Fig. 1c and d, light green colored violins) – indicating less reducing conditions –, compared to Koljö Fjord with higher sedimentary Mo and U enrichments (Fig. 1c and d, pink colored violins) – indicating more reducing conditions. Several factors controlling oxygenation in both fjords have been proposed, including limited water mass exchange related to bathymetrical characteristics (e.g., presence of sills, and narrow and deep basins), human activities (e.g., eutrophication), and natural variability (e.g., atmospheric drivers).

These will be discussed in the following sections.

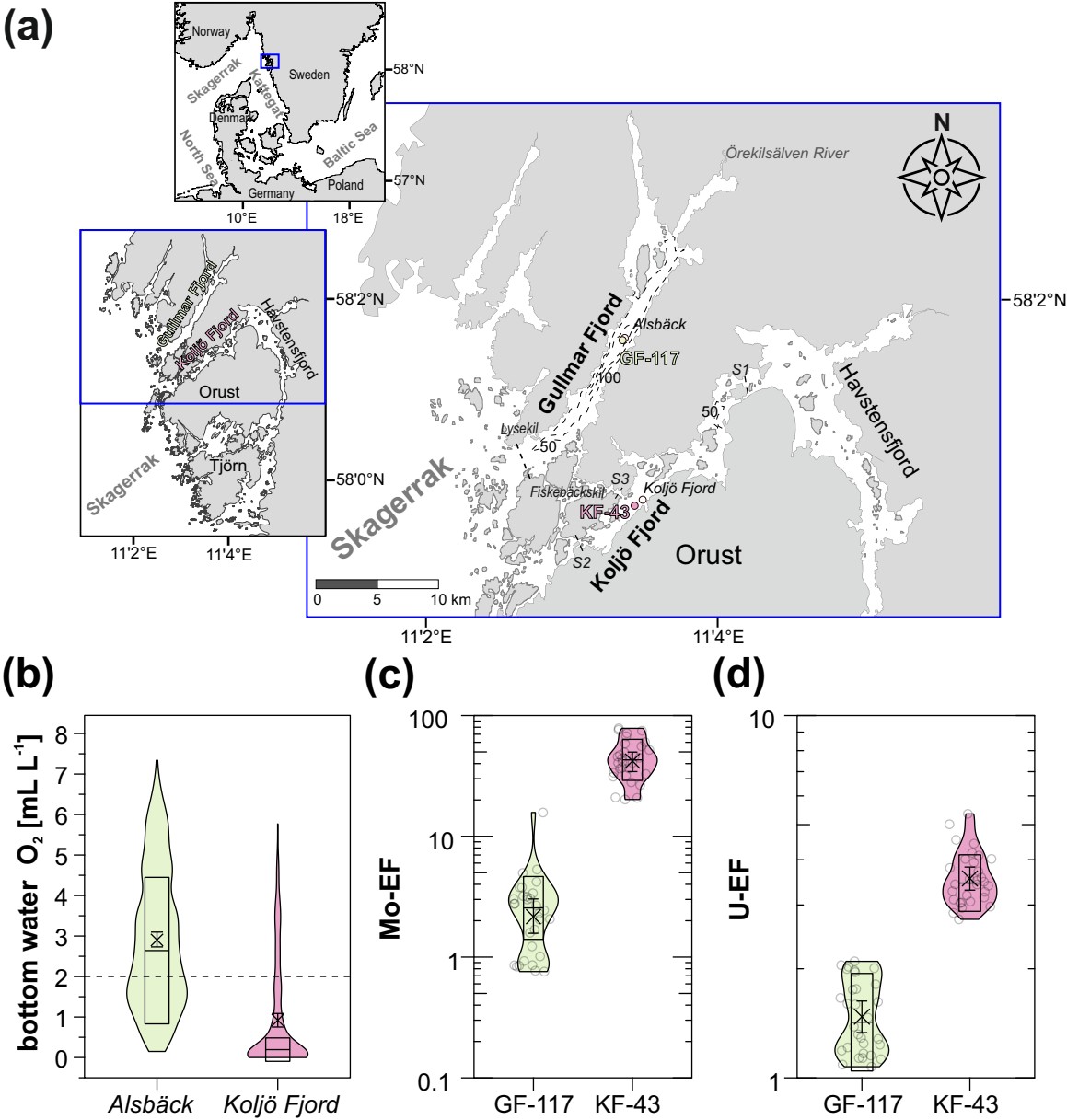

**Figure 1.** (a) Bathymetric map of the study area with the sites in Gullmar Fjord (GF-117, 58º 19.689'N, 11º 33.135'E, 115 m water depth) and Koljö Fjord (KF-43, 58º 13.624'N, 11º 34.265'E, 41.5 m water depth). Monitoring sites near both study sites "Alsbäck" (Alsbäck Deep, Gullmar Fjord, ~120 m water depth) and "Koljö Fjord" (Koljö Fjord, ~43 m
water depth) are indicated by an empty circle next to the sampling locations. Map adapted from Brinkmann et al. (2023a). (b) Ranges in bottom water $O_2$ at the two monitoring sites from 1951–2018 (obtained from the publicly available Svenskt HavsARKiv (SHARK) database, https://sharkweb.smhi.se/hamta-data/, provided by the Swedish Meteorological and Hydrological Institute, SMHI, 2022, last accessed: 03 September 2022), displayed as violin plots. The mean is indicated by a cross, and the 0.99 confidence interval (CI) of the mean by an error bar. The box within each
violin plot represents the median absolute deviation (MAD) and horizontal line in each box shows the median. The dashed horizontal line marks the upper dysoxic threshold at 2 mL $L^{-1}$ (Algeo and Li, 2020). The colors of the violin plots correspond to the color scheme used in Paul et al. (2023), illustrating the two "redox bins" 4 (ir/regularly dysoxic, light green) and 2 (ir/regularly euxinic, pink). (c) Molybdenum (Mo) enrichment factor (EF) and (d) Uranium (U) EF at both study locations from the September 2018 sampling campaign (original data and description of EF calculations are
outlined in Paul et al., 2023). Color coding, and statistical features as in (b). Individual observations are shown (empty circles within each violin plot).

### 2.2.1 Hydrographic characteristics controlling water mass renewal and deoxygenation

At Gullmar Fjord, fresh water enters from the Örekilsälven River and marine water from the Kattegat and Skagerrak to

which Gullmar Fjord opens across a sill at 42 m water depth (Harland et al., 2006, Fig. 1a). The mixed inflow of fresh

and marine water results in a strong and persistent thermohaline stratification. Brackish surface water (24–27) and almost fully marine salinities in the deep water (34–35, Nordberg et al., 2001; Arneborg, 2004) are separated by a variable pycnocline between 15–20 m below sea surface (Svansson, 1984). Deep waters are renewed and reoxygenated annually (usually in late NH winter or spring; Nordberg et al., 2000). Strength and duration of bottom water renewal and reoxygenation at Gullmar Fjord is driven by variability in the predominant wind direction and forcing, partially controlled by the North Atlantic Oscillation (NAO: normalized pressure differences between the Azores high and the Icelandic low; Hurrell, 1995; Chen and Hellstrom, 1999, Björk and Nordberg, 2003). When prevailing northeasterly–easterly winds are dominant over Scandinavia, the fjord is more oxygenated due to upwelling of highly saline Skaggerak deep waters along the Swedish coast, which ventilate the fjord (Rydberg, 1975; Harland et al., 2006). Conversely, when westerly winds are dominant, the fjord is less oxygenated due to a downwelling regime along the coastline. Annual bottom water exchange at Gullmar Fjord via the 42 m deep sill between Lysekil and Fiskebäckskil (Fig. 1a) leads to strong seasonal variability in bottom water $O_2$. Despite seasonally low $O_2$ levels and episodic dysoxia, $\Sigma H_2S$ has never been detected in the fjord during the past 70 years (Fig. 2, upper $O_2$ panel; Filipsson and Nordberg, 2004b).

The hydrography of Koljö Fjord (and the other fjords in the Orust and Tjörn island systemfig 1a) is dominated by brackish Kattegat–Skagerrak surface water originating from the Baltic Sea, while freshwater input is of minor importance as no major river discharges into the fjord (Björk et al., 2000, Filipsson et al., 2005). Mixing of the brackish surface waters (15–27) and more saline deep water (27–30), as well as deep water renewals, are less frequent and effective at Koljö Fjord compared to Gullmar Fjord. A strong pycnocline (between 15–25 m water depth) and shallower sill depths prevent direct inflow of deep waters from Skagerrak (Gustafsson and Nordberg, 1999; Filipsson and Nordberg, 2004a). Instead, deep waters enter via the adjacent Havsten Fjord, across a deeper sill between Havsten Fjord and Skagerrak at 20 m depth, before they reach Koljö Fjord (Harland et al., 2004). Deep waters at Koljö Fjord are renewed, but not necessarily reoxygenated, with a variable frequency (from annual to several years, Gustafsson and Nordberg, 1999). This has led to intermittently euxinic water masses between 15–20 m, typically during fall and winter, at least since the 1960s (Fig. 2, lower $O_2/\Sigma H_2S$ panel) – potentially already longer, although no monitoring data is available before this date (Rosenberg, 1990). The strength and occurrence of low $O_2$ conditions appears to be inverted between both fjords, despite their close proximity and connection to each other (sill S3, Fig. 1a). Shallow sill depths and dense, saline deep waters prevent inflow of saline and $O_2$-rich deep water into Koljö Fjord when environmental conditions favor re-oxygenation at Gullmar Fjord (cold winters and warm summers). Only when bottom water salinity has decreased to a level (<28.5), below which thermohaline stratification is weakened, ventilation is re-enabled

(typically at mild, humid winters with limited ice cover and cold summers; Björk and Nordberg, 2003; Nordberg et al.,

2001). At both fjords, natural processes, such as deep water exchange related to the fjord's bathymetry and

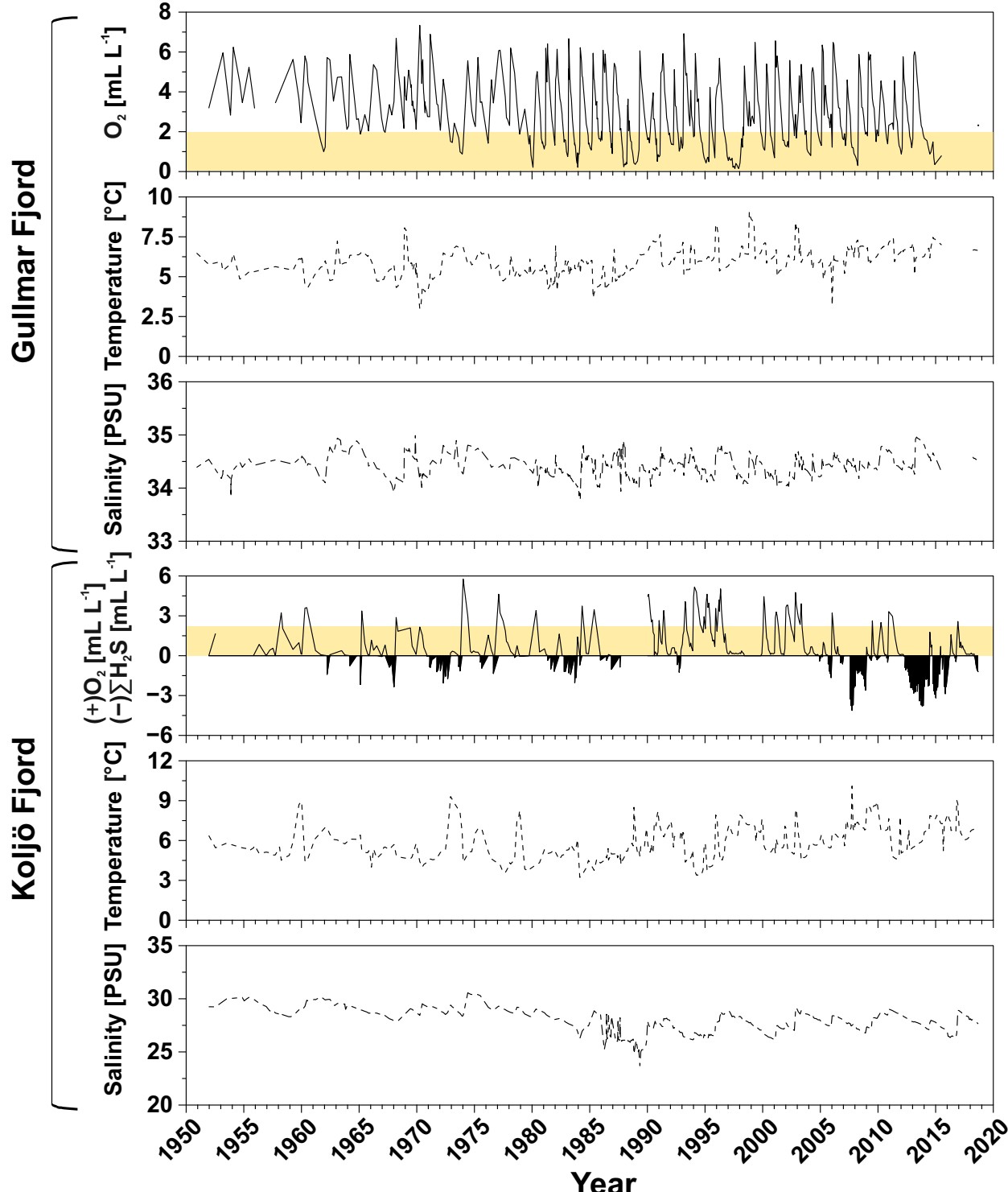

**Figure 2.** Bottom water monitoring data for Gullmar Fjord (upper three panels: $O_2$, temperature, and salinity) and Koljö Fjord (lower three panels: $O_2$ – positive values, $\sum H_2S$ – expressed as negative $O_2$, temperature, and salinity), recorded between 1950 and 2018 (SMHI, 2022). The yellow bars in the two $O_2$ panels indicate the dysoxic minimum and
155 maximum boundary (0.2–2 mL L$^{-1}$).

hydrography, and weather variability (wind strength and direction) play a large role in controlling environmental changes and deoxygenation (Nordberg et al., 2001; Filipsson and Nordberg, 2004b). **Figure 2.** (a) Bathymetric map of the study area with the sites in Gullmar Fjord (GF-117, 58º 19.689'N, 11º 33.135'E, 115 m water depth) and Koljö Fjord (KF-43, 58º 13.624'N, 11º 34.265'E, 41.5 m water depth). Monitoring sites near both study sites "Alsbäck" (Alsbäck Deep,

### 2.2.2 Sampling and sample processing

Sediment cores were collected aboard R/V Skagerrak during September 2018. Prior to core recovery, dissolved $O_2$, temperature, and salinity profiles of the water column were recorded by CTD (Brinkmann et al., 2022). From each site, GF-117 (115 m water depth) and KF-43 (41.5 m water depth), one set of duplicate cores were taken using a GEMAX[TM] twin-barrel short gravity corer (modified Gemini corer, 9 cm internal diameter, core length 20–60 cm, from Oy Kart AB, Finland). The duplicate cores were sampled for bottom- and pore water, and solid-phase geochemical analyses. Two series of bottom- and pore water were collected using Rhizons[TM] (pore size 0.12–0.18 μm) at 2 cm vertical resolution. The first series was collected for elemental analyses, and the second series for pore water $\Sigma H_2S$ analysis. The samples were collected into 10 mL polyethylene syringes through predrilled holes (diameter 4 mm, e.g., Jokinen et al., 2020a) immediately after core retrieval. The syringes for the $\Sigma H_2S$ analysis were prefilled with 1 mL of 10% zinc acetate solution to precipitate the $\Sigma H_2S$ as zinc sulfide (ZnS; Jilbert et al., 2018). Samples for elemental analyses were transferred into 15 mL polypropylene centrifuge tubes and acidified with 1M $HNO_3$ and stored dark at 4°C until further analysis.

Cores for solid-phase analyses were sliced at 0.4–2.0 cm intervals on deck immediately after core retrieval. Each sediment slice was transferred into a plastic bag, which was submerged into water to remove the remaining air, sealed, and then transferred into gas tight glass jars. To prevent oxidation artifacts (Kraal et al., 2009), the jars were flushed with nitrogen ($N_2$) and stored in a dark environment at -20°C until subsampling for the sequential extraction procedure. Subsampling of wet sediment samples was conducted under strict oxygen-free conditions inside a $N_2$-flushed glove bag. Each subsample was refrozen for 24h at -20°C and subsequently freeze-dried under vacuum for 48h prior to sequential extraction. Freeze-dried samples (instead of wet samples) were chosen for simultaneous and accurate determination of water and salt contents, porosity, and trace metal speciation on the same sample.

For estimating the water [g] and salt contents [g], and porosity [$cm^3$ $cm^{-3}$], each sample was pulverized and homogenized, and weighed in between each step using the bottom water salinity and the assumed solid-phase density of 2.65 g $cm^{-3}$ (Burdige, 2006). The gravimetric water content and salinity were used to determine the salt-free weight of the dry sediment to correct the solid-phase elemental concentrations for dilution by salt.

### 2.3 Pore water analyses

### 2.3.1 Major and trace elemental concentrations

The acidified bottom-, and pore water samples from both fjords were analyzed for Mo and U by Inductively Coupled Plasma-Mass Spectrometry (ICP-MS, Thermo Scientific XSeries 2, Department of Earth Sciences, Utrecht University) and for Al, Mn, Fe, and S by Inductively Coupled Plasma-Optical Emission Spectrometry (ICP-OES, Thermo Scientific iCAP 6000, Faculty of Forestry and Agriculture, University of Helsinki). Dissolved Fe and Mn are considered to be present as $Fe^{2+}$ and $Mn^{2+}$, although, some $Mn^{3+}$ (Madison et al., 2013) or colloidal and nanoparticulate Fe and Mn might also be present (Boyd and Ellwood, 2010; Raiswell and Canfield, 2012). Due to acidification of the pore water samples causing the release of $\Sigma H_2S$, dissolved S is assumed to be present primarily as sulfate ($SO_4^{2}$; Jilbert and Slomp, 2013). Pore water $\Sigma H_2S$ contents were analyzed spectrophotometrically (670 nm). This method is based on the dissolution of the ZnS precipitate and subsequent quantitative complexation of S as methylene blue (Jilbert et al., 2018). Measurements were calibrated with a series of standard solutions of hydrated sodium thiosulfate ($Na_2S_2O_3$ x $5H_2O$). Subsequently, the stock solution of $Na_2S_2O_3$ x $5H_2O$ was back titrated to determine the exact concentration of S in the solution (Burton et al., 2008).

### 2.3.2 Diffusive flux calculations

The diffusive fluxes of Mo and U ($F_{Diff}$) were determined using Fick's first law of diffusion (Eq. 1, Boudreau, 1997):

$$F_{Diff} = - \varphi(0) D_S \frac{\partial c}{\partial x} \tag{1}$$

where $\varphi(0)$ is the porosity of the surface sediment, $D_S$ is the molecular diffusion coefficient near the sediment water interface (SWI), $\partial c/\partial x$ denotes the concentration gradient between the bottom water and the uppermost pore water sample. Ds was determined from the seawater diffusion coefficient $D_{SW}$ (Eq. 2). Values for $D_{SW}$ for Mo and U were obtained from Li and Gregory (1974), following Morford et al. (2009), who assumed the diffusion coefficient of U to approximate that of Mo, rather than the value for the $UO_2^+$ complex. Based on the Stokes–Einstein relationship, $D_{SW}$ was corrected for ambient temperature, salinity, and pressure using an extended version of the *diffcoeff* function (Sulu-Gambari et al., 2017) in the R package *marelac* (v. 2.1.10) (Soetaert et al., 2010). Pore water salinity and temperature were assumed to equal the deepest bottom water value determined by the CTD. Subsequently, Ds was corrected for tortuosity (Eq. 2; Boudreau, 1997).

$$D_s = \frac{D_{SW}}{1-\ln(\varphi^2)} \tag{2}$$

## 2.4 Solid-phase analyses

### 2.4.1 Sequential trace metal extraction

Aliquots of ~100 mg freeze-dried sediment were used for solid-phase fractionation using a combination of different extraction methods (Table 1), based closely on Jokinen et al. (2020a). We acknowledge the long-standing debate about the validity of using freeze-dried vs. wet sediments for sequential extraction of trace metals (e.g., Kersten and Förstner, 1986; Hjorth, 2004). However, Jokinen et al. (2020a,b) observed no evidence for remobilization of highly redox-sensitive elements such as arsenic (As), as discussed by Huang et al. (2015), and concluded that their Mo and U data

were reliable. By following the same sample handling measures as Jokinen et al. (2020a,b), we consider the potential for significant introduction of artifacts due to freeze-drying low. The Al, Ca, Fe, Mn, Mo, S, and U contents were fractionated in the following pools: F1 – weakly-sorbed metal (Me) species; F2 – carbonates, acid volatile sulfur (AVS), Mn(II) phosphates, and labile Me -OM complexes; F3 – Fe (oxy)(hydr)oxides, Mn (oxy)(hydr)oxides, and labile Me-OM complexes; F4 – refractory Me-OM complexes; F5 – pyrite; and F6 – silicates.

**Table 1.** Sequential extraction procedure for trace metals. Adapted from Jokinen et al. (2020a); an additional nitric acid step was added, designated as F5, in order to extract pyrite as described in Poulton and Canfield (2005).

| Code | Fraction | Solvent | Time | Targeted minerals phase | References |
|------|----------|---------|------|------------------------|------------|
| F1 | Exchangeable | MgCl$_2$ (1 M), pH 8 | 0.5 h | Weakly-sorbed Me species | Tessier et al. (1979) |
| F2 | Acid-soluble | Na-acetate (1 M), pH 4.5 | 6 h | Carbonates<br>Fe monosulfide (FeS)<br>Mn(II) phosphates[a]<br>Labile Me-OM complexes<br>Labile Fe (oxy)(hydr)oxides (i.e., ferrihydrite and lepidocrocite) | Tessier et al. (1979)<br>Cornwell and Morse (1987)<br>Lenstra et al. (2021a)<br>Jilbert et al. (2018) |
| F3 | Reducible | Na-dithionite (5%),<br>Acetic acid (0.35 M),<br>Na-citrate (0.2 M), pH 4.8 | 4 h | Labile and crystalline Fe (oxy)(hydr)oxides (i.e., ferrihydrite and lepidocrocite, goethite, and hematite)<br>Mn (oxy)(hydr)oxides<br>Labile Me-OM complexes | Poulton and Canfield (2005)<br>Hermans et al. (2019b)<br>Lalonde et al. (2012) |
| F4 | Organic | Ashing at 550 °C<br>HCl (1 M), pH 0 | 2 h<br>24 h | Refractory Me-OM complexes | Ruttenberg (1992) |
| F5 | Strong-acid-soluble | HNO$_3$ (65-70%)<br>MQ wash step[2] | 2 h<br>0.5 h | Pyrite (FeS$_2$) | Claff et al. (2010) |
| F6 | Residual | HF (40%)<br>HClO$_4$:HNO$_3$ (3:2 vol%) | O/N[b] | Silicates | Poulton and Canfield (2005) |
| | Total | HF (40%)<br>HClO$_4$:HNO$_3$ (3:2 vol%) | O/N[c] | Sum of all phases | |

[a]assumed based on the extraction protocol from Lenstra et al. (2021a), who used ascorbic acid to extract Mn(II) phosphates. [b]O/N = overnight (~12 h). [c]An additional wash step was introduced here to remove residual concentrated nitric acid from the sample tubes.

All solution and reagents used in the sequential extraction were ultrapure for trace metal analysis and prepared with milli-Q water to avoid contamination. Since the mineral phases extracted in F1–F3 are very redox-sensitive, solvents used in these steps were purged with $N_2$ for 30 min prior to extraction (CDB solutions for F3 were only purged prior to adding Na-dithionite, otherwise purging would lead to loss of volatile S compounds, which are required to reductively dissolve metal oxides) and solvents were added to the samples under constant $N_2$ gas flow (F1–F3).

The Al, Ca, Fe, Mn, Mo, S, and U contents in the first five fractions were determined using ICP-MS (Agilent 7800 ICP-MS) at HelLabs (University of Helsinki). Blank corrections were applied to all extracts to correct for background contamination (Table S1 for details on the blank correction procedure). Additionally, Ca and S contents in F1 were corrected for any Ca or S associated with sea salt using the stochiometric ratio of seawater (Sverdrup et al., 1942) and pore water Ca and S data. Molybdenum contents could not be determined in F2, since data from almost all samples fell below the detection limit as determined from a 10-σ estimate of blanks of the Na acetate solution. Jokinen et al. (2020b) made a similar observation in their trace metal extraction protocol.

For determination of the elemental contents in the residual fraction F6, ~80 mg of the residual samples were microwave-digested at 200°C using 5 mL 65-70% $HNO_3$, 3 mL 34-37% $HCl_3$, and 3 mL 48% HF. After cooling at room temperature, the vessels were opened and 30 mL of 4 % boric acid was added to each vessel for HF neutralization. Afterwards, the vessels were closed tightly, placed into the microwave, and neutralized at 170°C. The sample digests were then analyzed for Al, Ca, Fe, Mn, Mo, S, and U concentration by ICP-MS (Triple QQQ ICP-MS) at HelLabs (University of Helsinki). Total contents of Al, Ca, Fe, Mn, Mo, S, and U were determined by the sum of all six fractions assuming that 100% is extracted.

### 2.4.2 Carbon and Nitrogen contents

Aliquots of ~ 0.25 g of freeze-dried sediment were decalcified using two wash-steps of 1 M HCl as described in Van Santvoort et al. (2002). After drying and re-powdering, the decalcified samples were analyzed for organic carbon ($C_{org}$) and total nitrogen contents on a LECO 2000 CNS analyzer (Ecosystems and Environment Research Programme, Helsinki University). The results were normalized against the international analytical standard sulfamethazine. The certified value for sulfamethazine is 51.8 wt. % for C and 20.1 wt. % for N. The obtained mean values for the analyses were 51.5 wt. % and 20.3 wt. % with a standard deviation of 0.4 wt. % and 0.2 wt. %, respectively. Average analytical uncertainty (relative standard deviation, RSD %) is based on sediment sample duplicates (n=3) was <4 wt. % for both C and N. For determination of the $C_{org}$ content in each sample, measured C and N contents were corrected for weight loss upon decalcification and salt content.

### 2.4.3 Organic matter source determination

Fjord systems receive OM loading from terrestrial ($OC_{terr}$) sources (e.g., plant material, soil, and weathering of bedrock), and marine biogenic ($OC_{phyt}$) sources (phytoplankton production, either autochthonous or allochthonous) (Smith et al., 2015; Prebble et al., 2018). A widely used tool to quantify the sources of OM in the aquatic environment is the C/N (or N/C) ratio (e.g., Thornton and Mcmanus, 1994; Wehrmann et al., 2014; Faust and Knies, 2019). Goñi et al. (2003). The contribution of $OC_{phyt}$ (Eq. 3) and $OC_{terr}$ (Eq. 4) to the total OM loading in estuary-type depositional environments can be approximately quantified using simple two-end-member mixing models, based on the molar N/C ratios of bulk OM (Goñi et al., 2003). Here, we use endmember values of $(N/C)_{terr}$ = 0.04 (terrestrial-C3 plant-derived), and $(N/C)_{phyt}$ =0.13 (riverine–estuarine phytoplankton), as per Jilbert et al. (2018).

$$\%OC_{phyt} = \frac{(N/C_{sample} - N/C_{terr})}{(N/C_{phyt} - N/C_{terr})} \times 100 \tag{3}$$

$$\%OC_{terr} = 100 - \%OC_{phyt} \tag{4}$$

We acknowledge that these end-member values approximate a large potential range of N/C values for both phytoplankton-derived and terrestrial OM. In this study, we report the results of the calculation including ranges as given for the maximum and minimum end-member combinations shown in the fields of Goñi et al. (2003), i.e., $(N/C)_{terr.}$ = 0.02–0.05 and $(N/C)_{phyt.}$ = 0.13–0.17.

### 2.4.4 Calculation of authigenic Mo and U accumulation rates

To determine the authigenic Mo and U accumulation rates ($TM_{MAR}$; Eq. 7), first the mass accumulation rate (MAR; Eq. 5) and then the authigenic trace metal concentrations of Mo and U were calculated, here expressed as the excess trace metal concentration ($TM_{XS}$; Eq. 6).

$$MAR\ (g\ cm^{-2}\ yr^{-1}) = SAR \times \rho \times (1-\varphi) \tag{5}$$

where SAR is the mean sediment accumulation rate (cm $yr^{-1}$), $\rho$ is the dry bulk density (2.65 g $cm^{-3}$) of sediments and $\varphi$ is the mean sediment porosity ($cm^3\ cm^{-3}$) at each site.

$$TM_{XS} = TM_{sample} - ((TM/Al)_{standard}) \times Al_{sample} \tag{6}$$

where $TM_{sample}$ is the trace metal concentration in the sediment sample, $Al_{sample}$ is the Al concentration in the sediment sample, and $TM/Al_{standard}$ is the ratio between the trace metal and Al in a standard, typically upper continental crust (UCC) values (Rudnick and Gao, 2014). Finally, Mo and U accumulation rates ($TM_{MAR}$, μmol $TM_{auth}$ $m^{-2}$ $yr^{-1}$) were estimated as:

$$TM_{MAR} = MAR \times TM_{XS} \times 10000 \tag{7}$$

where MAR is the mass accumulation rate (Eq. 5), $TM_{XS}$ ($\mu mol\ g^{-1}$) is the authigenic trace metal enrichment in each core in (Eq. 6), and 10000 is the conversion factor from $cm^{-2}$ to $m^{-2}$. A small set of $Mo_{XS}$ values (= 5 out of 30) in the middle section of the GF-117 core (10.625–23.75 cm) were negative, which were omitted from the MAR estimation.

**3 Age vs. depth model**

Age models for sediments at both coring locations are available from previous sampling campaigns conducted between 1996 and 2001, using $^{210}Pb$ dating and applying the constant rate of supply (CRS) model (Nordberg et al., 2000, Nordberg et al., 2001, Asteman et al., 2018), and biostratigraphy (Filipsson and Nordberg, 2004b). According to these studies, sedimentation rates for Gullmar Fjord are more difficult to estimate than for Koljö Fjord due to possible bioturbation artifacts. At Gullmar Fjord average sedimentation rates have been estimated to fluctuate between ~0.70 and ~0.90 cm $yr^{-1}$. To account for compaction and bioturbation, previous studies have assumed ~0.90 cm $yr^{-1}$ for the upper 15 cm and ~0.70 cm $yr^{-1}$ for the remainder of the sediment core at Gullmar Fjord (Filipsson and Nordberg, 2004b, Asteman et al., 2018). At Koljö Fjord, a previous study assumed a sedimentation rate of ~0.40 cm $yr^{-1}$ for the upper 25 cm and ~0.24 cm $yr^{-1}$ for the remainder of the sediment core (Nordberg et al., 2001). Sediment $C_{org}$ profiles at our sampling location in Koljö Fjord are comparable between sampling campaigns over recent decades and show distinct fluctuations around a value of ~6 wt. %, related to salinity variations (> or < 28–29, respectively) and the presence of laminae or lack thereof (Nordberg et al., 2001). Filipsson and Nordberg (2004a) used these relationships to establish an age model for sediment profiles from different Koljö Fjord sampling sites.

For developing our own age models, we assume similar sedimentation rates at each site to those reported previously, as a starting point. In case of Koljö Fjord, we further refine the sedimentation rate-based age model using the $C_{org}$-based age model of the sediment core collected in 1998, K6A (same location as KF-43). Tuning of $C_{org}$ content versus sediment depth was performed in the time-series tuning and analysis program *QAnalyseries 1.5.1 Win* (Kotov and Paelike, 2018) – a development of *AnalySeries 1.1.0* (Paillard et al., 1996). For dated $C_{org}$ sediment profiles of K6A and KF-43 we refer to the Supplementary Materials (Fig. S2a). Since no ages were available from the K6A sediment core for the top 0.5–8.5 cm (8.5 cm ≈ 1998 in KF-43), the assumed average sedimentation rate of ~0.40 cm $yr^{-1}$ for the upper 25 cm was used construct the age model for this interval. According to our sedimentation rate-based age model for Gullmar Fjord, the sediment core from site GF-117 covers approximately the last 80 years (0–59 cm, Fig. S2b). Using the combined sedimentation rate- and $C_{org}$-based age model for Koljö Fjord sediment core KF-43, we estimate an approximate time coverage of the last 160 years (0–49 cm, Fig. S2b).

## 4 Results

### 4.1 Pore water geochemistry

At the ir/regularly dysoxic Gullmar Fjord, $SO_4^{2-}$ remains relatively constant throughout the sediment core (~22–24 mM) – close to the modern global seawater value of ~28 mM –, while $\Sigma H_2S$ remains below detection limit (Fig. 3a). Dissolved Mn rapidly increases below the SWI up to 288 µM at 8.5 cm depth, followed by a gradual decrease down to ~110 µM at depth and simultaneous moderate release of dissolved $Fe^{2+}$, reaching a maximum of ~26 µM at 16.5 cm depth. Dissolved Mo ($Mo_{diss}$) primarily follows $Mn^{2+}$ with depth, and once $Fe^{2+}$ release commences, Mo follows $Fe^{2+}$ with depth. Local maximum Mo concentrations (230 and ~240 nM) are found at 8.5 and 14.5 cm, respectively – coinciding with $Mn^{2+}$ and $Fe^{2+}$ peaks. At depth, Mo gradually decreases, except for a third, smaller peak, which is also visible in $Fe^{2+}$ at 34.5 cm depth. This specific pattern is also visible in the dissolved U ($U_{diss}$) profile; the remaining profile, however, shows a somewhat diverging trend relative to Mo. Besides an initial decrease below the SWI, the U

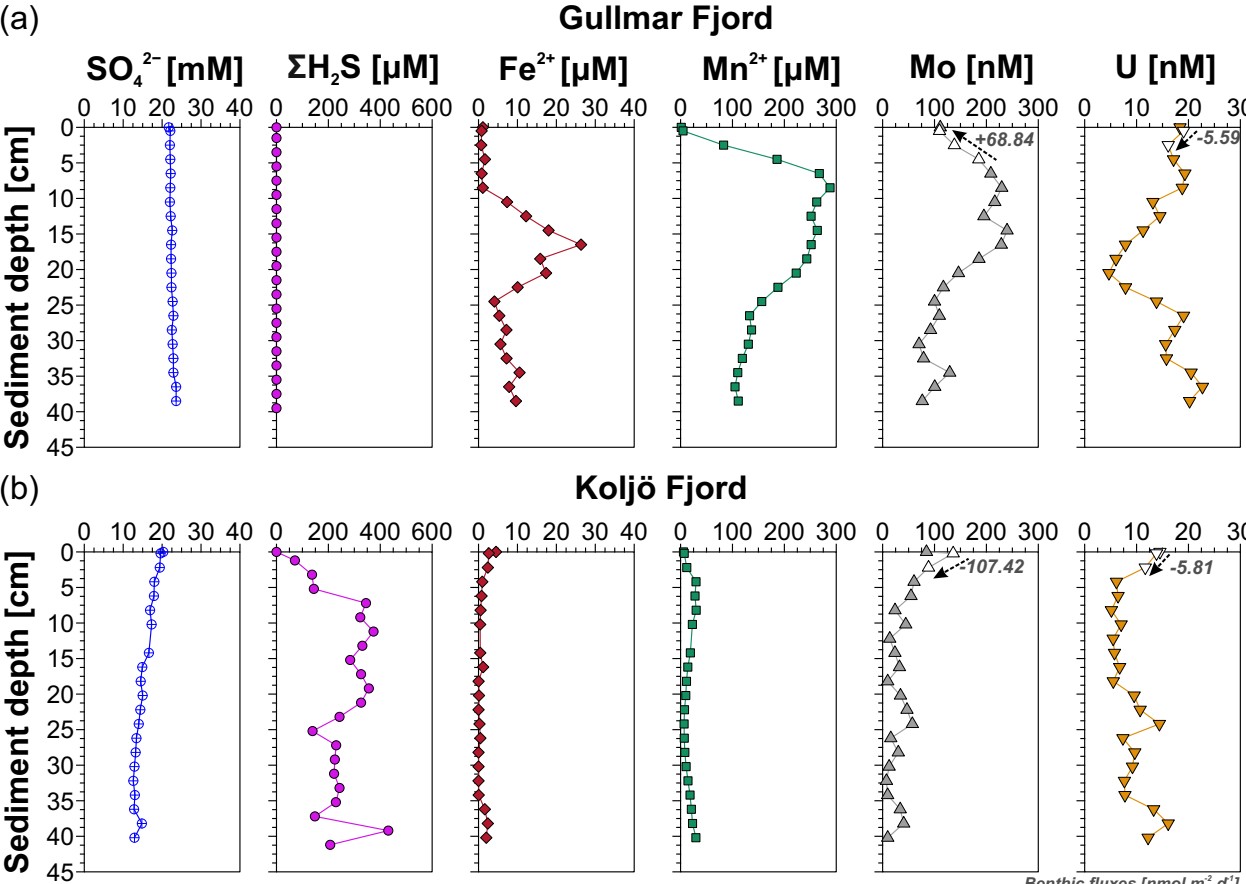

**Figure 3.** (a) Gullmar- and (b) Koljö Fjord downcore pore water profiles of major and trace pore water constituents: sulfate ($SO_4^{2-}$, $\Sigma H_2S$, Fe, Mn, Mo, and U. Estimated benthic fluxes (in nmol m$^{-2}$ d$^{-1}$) for Mo and U are provided. Positive benthic fluxes (+, upward facing arrow) refer to benthic release and negative fluxes (−, downward facing arrow) refer to sedimentary uptake. The white filled symbols indicate the samples used for the benthic flux estimation. Bottom water values not following a linear gradient with surface sediment values were omitted from the calculation (i.e., Mo at both fjords, and U at Gullmar Fjord).

maxima in the upper sediment column shows a slight offset to Mo maxima at 6.5 cm (~19 nM), and 12.5 cm (~14.5 nM), respectively. Furthermore, while Mo decreases in the deeper sediment, U increases below its minimum at 20.5 cm (~5 nM) and continues to increase at depth reaching two local maxima at 26.5 cm (~19 nM), and 36.5 cm (~23 nM), respectively. Overall, U concentration is tenfold lower than that of Mo.

At the ir/regularly euxinic Koljö Fjord, $SO_4^{2-}$ gradually decreases with depth from ~20 mM down to ~13 mM, while $\Sigma H_2S$ rapidly increases below the SWI to ~370 µM at 11 cm depth and remains high between ~200–400 µM throughout the sediment core (Fig. 3b). The linear gradient in the pore water shows an $\Sigma H_2S$ efflux into the water column (582 µmol m$^{-2}$ d$^{-1}$) and an influx of $SO_4^{2-}$ into the sediment (-18 mmol m$^{-2}$ d$^{-1}$). The strongly reducing character of the pore water is further illustrated by a sharp drop in $Fe^{2+}$ at the SWI (~2–4 µM), below which $Fe^{2+}$ remains extremely low with depth. Moreover, $Mn^{2+}$ is tenfold lower compared to Gullmar Fjord; highest concentrations are found between 2.5 and 15 cm (~12–30 µM), and below 30 cm (~14–30 µM), respectively. Molybdenum and U profiles resemble each other more closely compared to Gullmar Fjord, showing at least three distinct coinciding peaks: below the SWI, at 24 cm, and 28 cm depth.

**4.2 Solid-phase geochemistry**

Solid-phase geochemistry strongly differs between the two fjords (Fig. 4). While $C_{org}$ in Koljö Fjord is two times greater than in Gullmar Fjord, both fjords have similar molar N/C ratios of 0.08–0.11 (or as molar C/N ratio: 8.7–13.3) throughout the sediment cores. These ranges plot between phytoplankton and terrestrial derived OM (Bordovskiy, 1965; Meyers, 1994; Goñi et al., 2003; Lamb et al., 2006), as expected from fjord settings. Based on the two end-member mixing model, in both fjords the contributions of $OC_{terr}:OC_{phyt}$ to the total $C_{org}$ loading are ~40%:60% with an absolute range of ±7–24% around these values, depending on the end-members chosen (section 2.4.3.). We also note the possibility of diagenetic alteration of sediment C/N ratios influencing the estimates (e.g., Van Mooy et al., 2002), although this effect likely falls within the error ranges of the end-members.

**Gullmar Fjord**

*Manganese*

Total Mn contents correspond to those from previous studies on Gullmar Fjord sediments (Engström et al., 2005; Goldberg et al., 2012). Highest Mn enrichments are found in the surface sediments (14 to 399 µmol g$^{-1}$ ≡ 0.1 to 2.2 wt.%, primarily in F3 as Mn (oxy)(hydr)oxides (Mn oxides hereafter), and secondary in F2 as Mn carbonates (Fig. 4a). Below 10 cm depth, Mn is almost exclusively associated with F2. Notably, there are two distinct Mn peaks at ~20 and ~40 cm depth. Fractions 1 (weakly sorbed metal species), 4 (OM complexes), 5 (pyrite bound), and 6 (residual phase, i.e., silicates) do not play a major role as host phases for Mn in this system.

*Iron*

While total Fe concentrations are two times greater than Mn (731 to 897 µmol g$^{-1}$ ≡ 4.1 to 5.0 wt.%, Fig. 4a), significant trends with depth in sediment are less pronounced in Fe compared to Mn. The shape of the total Fe profile is mostly impacted by F2 – Fe carbonates (i.e., siderite and ankerite) – with the co-occurance of two peaks at approximately the same depths as the Mn peaks. Fraction 3 – Fe (oxy)(hydr)oxides (Fe oxides hereafter) shows a modest downward decreasing trend after a ~ 7 cm thick subsurface peak. Other fractions are either missing (F1) or show negligible variation (F4–F6).

*Molybdenum*

Sedimentary Mo is strongly coupled to Mn cycling (Mn oxides) at the surface sediment in weakly-sorbed metal species (F1 and F3, Fig. 4a). Except for the massive surface Mo enrichment, Mo remains below 10 nmol g$^{-1}$ on average. Remarkably, the gradual increase of F1 and F3 below 24 cm depth cannot be linked to changes in Mn content. Overall, Mo and Mn appear to be decoupled at depth, because the F2 peaks in Mn do not match the Mo profile. Fraction F2 is entirely absent, in accordance with a previous study (Jokinen et al., 2020b). We note that this result is not a consequence of analytical challenges. Although the Na acetate matrix has a comparatively high detection limit for Mo (see section 2.5.1.), the value of this detection limit is equivalent to approximately 1 nmol Mo g$^{-1}$ sediment, a negligible value in comparison to the other fractions, hence the absence of Mo in F2 is considered genuine.

*Uranium*

Decoupling is also apparent for Mo and U to the extent that U is sequestered in completely different phases than Mo. Uranium is largely associated with the residual (silicate) fraction F6 (~53%), followed by similar proportions of carbonates (F2), and refractory OM complexes (F4, Fig. 4a). Both F4 and F6 do not show a signiciant trend with depth, while F2 increases with depth (particularly below the Mn F2 peak at 20 cm depth), accompanied by F1 and F3. These trends are similar to those visible in the Mo solid-phase data.

**Koljö Fjord**

*Manganese*

The solid-phase Mn content at Koljö Fjord is as much as 20-times lower compared to Gullmar Fjord (9 to 25 µmol g$^{-1}$ ≡ 0.05 to 0.14 wt.%, Fig. 4b). Moreover, the proportions between the six fractions are different: Mn is primarily enriched in F6, closely followed by F4, and total proportions of F1 and F2 make up only one-third of those of F4 and F6. All four fractions share distinct enrichment peaks, albeit none of these peaks are present in all fractions simultanously. Strikingly, only in F4 and F6, peaks occur below 30 cm, whereas the other fractions show similar concentrations. All Mn peaks are also present in the solid-phase Fe extraction data at the same depth intervals (F2–F6), albeit less

pronounced except for the two subsurface maxima. Manganese in F3 shows the same two peaks visible for Mn in F2, F4, and weakly in F6. However, given that F3 has the lowest total concentration of all fractions (< 1 nmol g$^{-1}$), it probably only plays a subordinate role for authigenic sequestration of Mn at Koljö Fjord. The same applies to F1, which

is not preserved under sulfidic conditions.

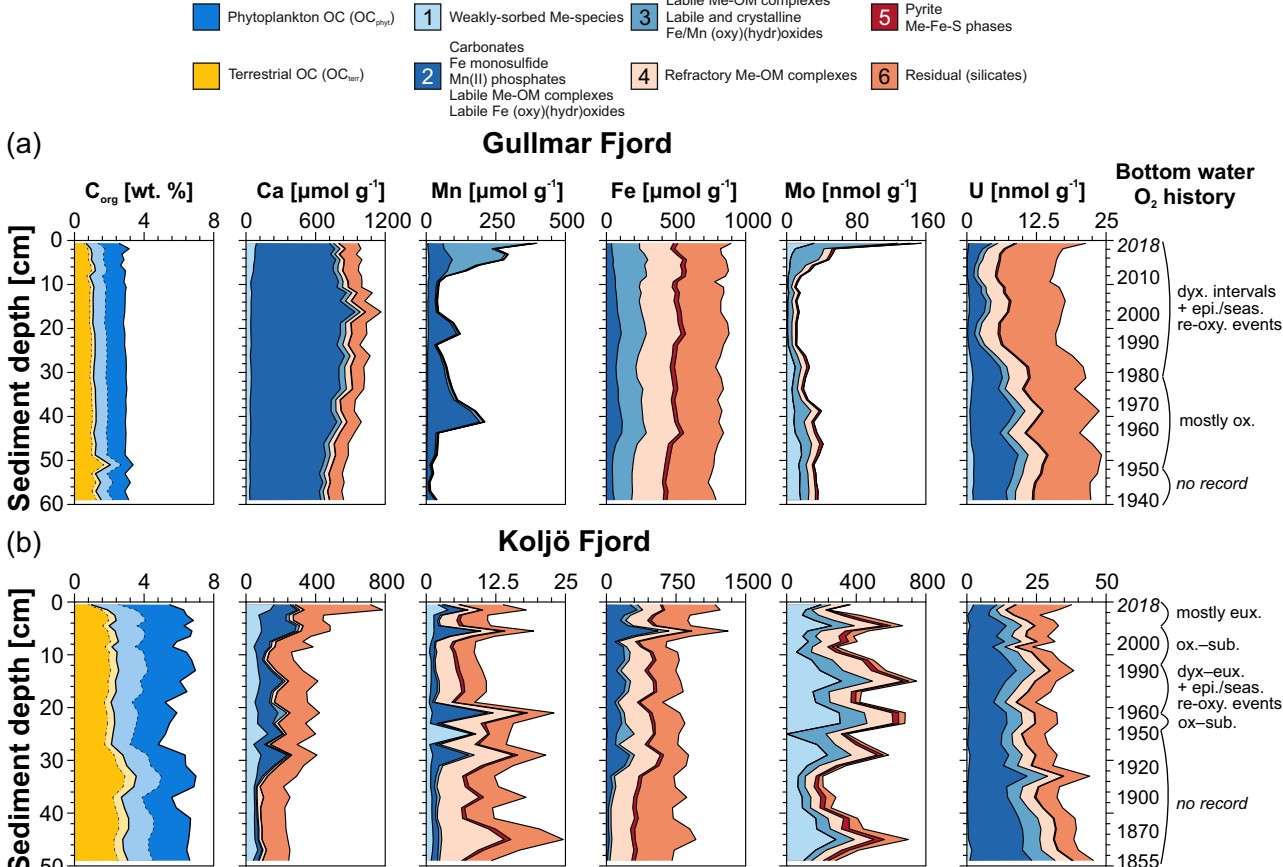

**Figure 4.** (a) Gullmar- and (b) Koljö Fjord downcore solid-phase distribution of $C_{org}$ – divided into riverine-estuarine-phytoplankton-derived (blue) and terrestrial-plant-derived (yellow), and elemental Ca, Mn, Fe, Mo, and U contents – divided into six different fractions (F1–F6) based on the sequential extraction scheme (Table 1). In each $C_{org}$ plot, the light-shaded area (light yellow, light blue) denotes the absolute range in $OC_{phy}$ and $OC_{terr}$ fractions based on the
maximum $(N/C)_{terr}$ and $(N/C)_{phyt.}$ endmembers (dashed line), and minimum $(N/C)_{terr}$ and $(N/C)_{phyt.}$ endmembers (dashed-dotted line), as reported in Goñi et al. (2003). The solid line denotes the $OC_{terr}:OC_{phyt.}$ contributions calculated when using the $(N/C)_{terr.} = 0.04$ and $(N/C)_{phyt.} = 0.13$ endmembers, as per Jilbert et al. (2018). A summary of bottom water $O_2$ history at both fjords is provided on the right. The bottom water redox conditions were derived from the fjord's monitoring data (Fig. 2, SMHI, 2022) and correlated to selected depth intervals in the sediment cores using the
estimated age-depth models (section 3, Fig. S2a+b). The abbreviations used to describe the average bottom water (BW) redox condition are defined as follows (in order of appearance top to bottom): dyx = dysoxic, ox = oxic, epi. = episodical, seas. = seasonal, re-oxy = re-oxygenation, eux = euxinic, and sub = suboxic).

*Iron*

Total Fe contents are slightly more elevated compared to Gullmar Fjord (607–1311 µmol g$^{-1}$ ≡ 3.4–7.3 wt.%; Fig. 4b).
By contrast to Mn, Fe in F4 and F6 constitute ~60% of the total Fe host phases. While F4 does no show any clear trend throughout the sediment core, the size of F6 shows a clear separation between strong enrichments at the top and at the bottom of the core, separated by a less enriched central part of the sediment core (≈ half of the top and bottom

enrichment). The third largest Fe pool consists of Fe carbonates (i.e., siderite or ankerite) or iron monosulfide (FeS) extracted in F2; however, below 30 cm, this pool dissapears.

*Molybdenum*

Molybdenum strongly follows Mn and Fe, showing enrichment maxima at similar sediment depths, albeit Mo peaks are more pronounced (particularly between 10 and 20 cm, and at 45 cm depth, respectively, Fig. 4b) and range between ~200 and 750 nmol g$^{-1}$. Molybdenum peaks are dominated by F1 followed by equal proportions of F3 and F4 – both likely sulfurized OM. Despite high pore water $\Sigma H_2S$, the pyrite pool F5 is unexpectedly low and of similar size as the residual fraction F6. Fraction 2 is also absent at Koljö Fjord (see section 4.2. Gullmar Fjord).

*Uranium*

Uranium partially covaries with Mo with respect to the occurrence of peaks (Fig. 4b). However, below 25 cm U gradually increases with depth with two peaks at 33 cm and at the core bottom (~49 cm), whereas Mo drops below 30 cm and only sharply increase again from 40–45 cm. Besides these differences, U host phases are contrasting those of Mo and differ from Gullmar Fjord – except for F1 and F5 being the smallest fraction. Approximately 40% of the U contents are associated with F2; the remaining 60% is divided among F6, to similar portions by F3 and F4, whereas F1 and F5 make up only 2 and 1% percent, respectively, of the total extractable U.

## 5 Discussion

### 5.1 Iron and Manganese cycling and sequestration mechanisms

Sedimentary Mn and Fe geochemical cycling, contents and speciation differ between Koljö- and Gullmar Fjord (Fig. 3 and 4). These differences are likely caused by distinct depositional environmental processes within the fjords. Such processes include riverine input and salinity-driven flocculation of Mn and Fe oxides during estuarine mixing processes at the freshwater–marine interface (Sholkovitz, 1978; Brinkmann et al., 2023b), shelf-to-basin shuttling (e.g., Lenz et al., 2015a; Lenstra et al., 2020; Lenstra et al., 2021b), gravitational focusing of suspended material (sedimentary Mn oxide enrichments are usually highest in the deepest part of a basin; Hermans et al., 2019b), subsequent "refluxing" (reductive dissolution–oxidative precipitation) of Mn and Fe oxides between the oxic/dysoxic redox interface (within the water column or the sediment; e.g., Adelson et al., 2001; Sulu-Gambari et al., 2017), and enhanced sequestration of Fe compared to Mn under suboxic and particularly sulfidic conditions (Hermans et al., 2019b). Occurrence, strength, and interaction between such processes is sensitive to temporal variability in seasonal changes in water mass stratification, lateral transport of sediments, and redox conditions (Lenz et al., 2015a; Sulu-Gambari et al., 2017; Scholz et al., 2019).

### 5.1.1 Manganese (Mn)

At Gullmar Fjord, sedimentary Mn mostly consists of non-silicate Mn species – Mn oxides (F3) and Mn carbonates (F2) (Goldberg et al., 2012). Enrichments peaks of Mn are found in the surface sediments, present as Mn oxides (Fig. 4a). Oxic water conditions, which prevailed prior to our sampling campaign (Fig. 2), likely stimulated precipitation of Mn oxides from dissolved $Mn^{2+}$ upon contact with $O_2$, and their subsequent shuttling to the seabed (Dellwig et al., 2018; Lenstra et al., 2021b). Upon reductive dissolution of Mn oxides, $Mn^{2+}$ is released to the pore water (Fig. 3a) and subsequently re-precipitates as Mn carbonates upon contact with bicarbonates (Calvert and Pedersen, 1996). Besides a strong subsurface maximum in $Mn^{2+}$ at Gullmar Fjord (Fig. 3a), dissolution of Mn oxides continues (albeit slower) deeper in the sediment within the zone of Fe(III) reduction (Fig. 3a, Goldberg et al., 2012), as evident by elevated $Mn^{2+}$ (~100 μM) at depth and the presence of two comparatively small Mn oxide peaks at ~20 and ~40 cm depth, respectively. At the same depth intervals, our data also show two Mn peaks in F2, which likely consist of Mn carbonates (Fig. S4) – although some Mn(II) phosphates might also be present (Hermans et al., 2019b; Hermans et al., 2021). Distinct layers of authigenic Mn carbonates are a common observation in dysoxic–suboxic sediments beneath oxic bottom waters in many coastal marine environments (e.g., Huckriede and Meischner, 1996; Lenz et al., 2015b; Lenstra et al., 2020). Formation of Mn carbonate enriched layers at Gullmar Fjord (and similar depositional environments) may either represent relic shifts in the redox boundary within the sediment (Goldberg et al., 2012) as described in Burdige (1993), or are a result of changes in Mn input related to varying intensity in Mn oxide shuttling – where stronger Mn oxide shuttling promotes Mn carbonate accumulation (e.g., Lenz et al., 2015a; Lenstra et al., 2021b).

At Koljö Fjord these processes also occur but are less efficient due to high sulfate reduction rates, inducing $\Sigma H_2S$ release to the pore water (Fig. 3b), which in turn alters the vertical zonation of electron acceptors used for OM degradation (Burdige, 1993). Under such sulfidic pore water conditions, Mn oxide formation is restricted to sufficiently oxygenated zones in the water column (Brewer and Spencer, 1971), which at Koljö Fjord occur above the pycnocline (~15–25m water depth; SMHI, 2022). When these Mn oxides sink through the suboxic water column, they begin to reductively dissolve before reaching the sediment (Burdige, 1993; Scholz et al., 2017). Only a small fraction of Mn oxides may survive the dissolution process in the water column, which is the case here, as inferred by the minor Mn oxide peak close to the sediment surface (~1.5 cm). Subsequently, these Mn oxides are available for rapid conversion into Mn carbonates in the sediment (Lenz et al., 2015a; Lenstra et al., 2021a). Coinciding Ca and Mn peaks in F2 throughout the sediment core, suggest that past fluctuation in the redox conditions have allowed conversion of Mn oxides to Mn carbonates. However, the more reducing water column and pore water conditions prevent a long-term

build-up of these two Mn host phases (Fig. 4b). This explains why the majority of sedimentary Mn resides in the silicate fraction (F6) and refractory OM complexes (F4) and the 20-times lower total Mn contents at Koljö Fjord compared to Gullmar Fjord, where more oxygenated conditions and deeper water depth promote Mn oxide formation and gravitational focusing on the surface sediments.

### 5.1.2 Iron (Fe)

Compared to Mn, total sedimentary Fe contents at Gullmar Fjord show hardly any trend with depth (Fig. 4a), analogous to estimates by Goldberg et al. (2012). The largest Fe fraction F6, probably represents Fe bound to illite, which is the most common clay mineral in this area (Hassellöv et al., 2001), and the smallest Fe fraction F5 is likley a residual from a previous phase rather than actual pyrite, as $\Sigma H_2S$ was below detection limit. Fractions F2 and F3 show the strongest variability among all fractions. As no separate extraction of labile Fe oxides was performed, F3 likely consists of a mixture of both labile (i.e., ferrihydrite and lepidocrocite) and crystalline (i.e., goethite and hematite) Fe oxides (Table 1). Based on experiments performed by Poulton and Canfield (2005), we cannot rule out that a minor fraction of labile Fe oxides (~1–2 %) may have already been extracted in the previous Na-acetate step (F2). With regards to F3, we observe a surface enrichment in the upper 10 cm, which is likely due to the formation of labile Fe oxides. This assumption is based on the Fe and Mn pore water profiles, showing that the F3 peak coincides with maximum dissolution of Mn oxides (release of $Mn^{2+}$), which itself catalyzes labile Fe oxide formation by oxidation of $Fe^{2+}$ (Fig. 3a and 4a; Wang and VanCappellen, 1996). With onset of reductive dissolution of Fe oxides below ~10 cm depth, these labile Fe oxides are being readily dissolved. Underpinned by the results of Goldberg et al. (2012), we therefore infer that below 10 cm, F3 mostly contains crystalline and refractory Fe oxides and to a lesser extent labile Fe oxides. Two distinct Fe peaks are present in F2 that overlap with the Mn peaks at ~20 and ~40 cm depth. Given the apparent low sulfate reduction rates and release of dissolved $\Sigma H_2S$ required for Fe monosulfide (FeS) formation (Canfield et al., 1993; Aller, 1994), we do not expect any FeS present in F2. Instead, Fe peaks likely consist of Fe carbonates, which typically co-precipitate with rhodochrosite and calcite (e.g., Wittkop et al., 2020). Correspondingly to the Mn carbonate peaks, those Fe carbonate peaks likely represent past maxima in Fe oxide accumulation in response to enhanced Mn and Fe oxide refluxing (e.g., Lohan and Bruland, 2008; Lenstra et al., 2021b).

Total Fe contents at Koljö Fjord are comparable to Gullmar Fjord, since more reducing conditions do not impede Fe sequestration (e.g.; Hermans et al., 2019b; Hermans et al., 2021) – as it is the case for Mn (section 5.1.1.). The contrasting redox conditions, however, impact the type of Fe host phases present in the sediment. At Koljö Fjord, Fe is dominantly sequestered in F4 and F6, followed by F2, and thereby follows the distribution of Mn. Other similarities to Mn are the number of distinct peaks and their occurrence in the sediment profile, and the disappearance of fractions F1–

F3 below 30 cm. These patterns suggest a common control impacting the geochemical sedimentary cycling of both metals. Analogous to Mn, Fe oxide formation requires oxidizing conditions, unlike Mn oxides, however, crystalline Fe oxides (F3) can be preserved under sulfidic conditions (Hermans et al., 2021; Lenstra et al., 2021b), which explains their presence throughout the sediment core, although some Fe might also be sorbed to OM extracted in the same phase (Lalonde et al., 2012; Jokinen et al., 2020a). Whereas dissolved $Mn^{2+}$ does not commonly precipitate as Mn sulfides (MnS) upon contact with pore water $\Sigma H_2S$ (Suess, 1979; Carman and Rahm, 1997), dissolved $Fe^{2+}$ typically removes $\Sigma H_2S$ from the porewater, leading to the precipitation of FeS, which is extracted in F2 (Berner, 1980; Burdige, 1993). In agreement with co-variation patterns between Fe and Ca, F2 may also contain Fe carbonates, or labile Fe-OM complexes (Jilbert et al., 2018; Fig. 4). During diagenesis, FeS may be further transformed into pyrite (F5, Boesen and Postma, 1988). Based on the Fe:S ratio in F5 and the onset of $\Sigma H_2S$ release to the pore water (Fig. 3b), we suspect that below the subsurface Fe peak in F2 (~1.5 cm) at least part of F5 represents pyrite (Fe:S ratio $\geq 0.5$; Fig. S3).

## 5.2 Molybdenum and Uranium speciation and sequestration mechanisms

Total authigenic sedimentary enrichments of Mo and U are greater at Koljö Fjord compared to Gullmar Fjord (Fig. 1c,d and 4), which agrees with their geochemical redox behavior under euxinic and dysoxic conditions, respectively (Bennett and Canfield, 2020; Paul et al., 2023). Moreover, molar Mo/U bottom water and surface sediment ratios are elevated at both fjords (Fig. S4) with respect to the molar Mo/U ratio in average seawater (~7.5, Algeo and Tribovillard, 2009), indicating a greater mobility and subsequent sequestration of Mo compared to U (Scholz et al., 2013). Indeed, our sequential extraction data reveal greater authigenic Mo sequestration at both fjords relative to U (Fig. 1c, d and 4).

### 5.2.1 Molybdenum and Uranium in Gullmar Fjord

**Molybdenum**

At Gullmar Fjord, the molar Mo/U ratio in the pore water is >7.5 and gradually rising until 20 cm sediment depth (Fig. S4), which is consistent with the extraction data showing a dominance of Mo and Mn co-enrichments (F3) over U in the upper 10 cm. This distinct Mo-Mn covariation is likely caused by Fe and Mn oxide shuttling from the water column to the surface sediment, favored by ambient water column redox conditions. Under such conditions, Mo (as molybdate, $MoO_4^{2-}$) has a strong affinity to Mn- and Fe oxides. By attaching to these oxides, Mo is removed from the water column and "shuttled" to the sediment surface (Berrang and Grill, 1974; Scholz et al., 2013; Dellwig et al., 2021). As $O_2$ is consumed within the upper 2–5 mm of the sediment (Brinkmann et al., 2023b) – which is typical for non or mildly bioturbated coastal marine sediments underlying a relatively oxygenated water column (Glud et al., 2003, Slomp et al., 2013; Hermans et al., 2019a) – first Mn, and then Fe oxides are being reductively dissolved, by which Mo is

subsequently released to the pore water (Fig. 3a, e.g., Sulu-Gambari et al., 2017). A comparison between Mo accumulation rates ($Mo_{MAR}$) of the upper 10 cm and Mo diffusive fluxes ($Mo_{benthic\ flux}$), reveals that most of the released Mo will likely be buried in the sediment. In this low-sulfide system and thus in the absence of sulfide-mediated pathways for authigenic Mo sequestration, however, a fraction of $Mo_{diss}$ might diffuse upwards and escape burial, indicated by a positive $Mo_{benthic\ flux}$ (Table 2). This is a common mechanism observed in coastal depositional environments subject to periodic or seasonal reoxygenation events, such as the Major Baltic Inflows (MBIs, e.g., Scholz et al., 2013; Dellwig et al., 2021) or in response to the collapse of seasonal water mass stratification (Sulu-Gambari et al., 2017).

With the dissolution of Mn and Fe oxides and subsequent conversion of such into Fe and Mn carbonates (F2), solid-phase Mo concentrations drop to background levels (below 10 cm, Fig. 4a). This observation highlights the importance of Mn oxides as carrier phases for Mo in surface sediments underlying seasonally dysoxic bottom water (e.g., Sulu-Gambari et al., 2017), but also that more reducing conditions are required to permanently sequester Mo in the sediment (e.g., Tribovillard et al., 2006; Jokinen et al., 2020b). Below 20 cm, however, solid-phase Mo concentrations gradually increase again with depth while pore water Mo concentrations show a decreasing trend. The solid-phase increase in Mo is mostly manifisted in F1 (almost 20-times higher than the background). A similar observation was made by Jokinen et al. (2020b) in sulfidic sediments in the Finnish Archipelago underlying an oxic water column. The authors infered that the occurance of Mo in F1 deeper in the sediment represent a precurser phase of Mo sequestration following the Fe sulfide pathway (Vorlicek et al., 2018; Helz and Vorlicek, 2019). In contrast to Jokinen et al. (2020b), the pore waters at Gullmar Fjord are depleted in $\Sigma H_2S$ (at least down to 40 cm depth), due removal of any produced $\Sigma H_2S$ by reactive Fe oxides in the sediment (Goldberg et al., 2012). In turn, thiolation of molybdate is strongly limited (Helz et al., 1996; Erickson and Helz, 2000). As molybdate is the more likely pore water species of Mo throughout the core (Goldberg et al., 2012), we suspect that most Mo is attached to Fe oxides or weakly-sorbed to other mineral phases.

**Uranium**

Uranium (as uranyl complex with mostly carbonate or phosphate; Langmuir, 1978) may be absorbed to Fe- and, to a lesser extent, to Mn oxides in surface sediments (McKee et al., 1987; Morford et al., 2007; Brennecka et al., 2011; Singh et al., 2012; Dang et al., 2016). However, shuttling-induced removal and sediment surface focusing, as evident for Mo, is believed to be a less important control for U sequestration in marine sediments (Dellwig et al., 2021). Instead, diffusion across the SWI is regarded as the primary transport pathway of U to the sediment (Barnes and Cochran, 1991; Klinkhammer and Palmer, 1991; Algeo and Maynard, 2004; McManus et al., 2005). Our $U_{MAR}$ and

benthic flux estimates suggest that, while diffusion across the SWI appears to drive U sequestration in the surface sediments (negative flux, Table 2), presence of an additional particulate flux is possible. This is inferred from MARs five times greater than the diffusive flux (Table 2). One possibility is the deposition of U as Particulate Non-Lithogenic U (PNU) associated with OM (Hirose and Sugimura, 1991; Zheng et al., 2002b). However, preservation of such PNU is unlikely at Gullmar Fjord due to surface water $O_2$ concentrations >200 µM. Alternatively, the additional particulate U flux originates from U adsorption to Fe- and (Mn) oxides close to the SWI and subsequent "shuttling" of U the sediment surface.

According to thermodynamics, reducing conditions are required for permanent sequestration of U as particle reactive U(IV) uraninite (Veeh, 1967; Bonatti et al., 1971; Anderson et al., 1989; Klinkhammer and Palmer, 1991). Besides, abiotic U(IV) reduction, U may also be reduced by biotic reactions in presence of dissimilatory metal-reducing bacteria and/or sulfate-reducing bacteria, leading to the formation of both crystalline uraninite (e.g., Bargar et al., 2008, Sharp et al., 2009, Lee et al., 2010) and monomeric (non-crystalline) non-uraninite (e.g., Lovley et al., 1991; 1993; Fredrickson et al., 2000; Bhattacharyya et al., 2017).

In contrast to Mo, pore water U does not increase with the onset of Mn oxide dissolution, but rather shows a decreasing trend (Fig. 3a). Coinciding with maximum Fe oxide dissolution ($Fe^{2+}$ release) this decreasing trend is amplified and continues until ~20 cm, at which most $Fe^{2+}$ has been consumed. The coinciding drawdown of pore water U with increasing $Fe^{2+}$ is a typical observation in both experimental and field data of marine sediments (Cochran et al., 1986; Klinkhammer and Palmer, 1991; Zheng et al., 2002a) and has been attributed to a combination of Fe and U reduction commencing at similar redox potentials and efficient removal of particle reactive U(IV) from pore water (Cochran et al., 1986; McKee et al., 1987; Zheng et al., 2002a). According to U reduction kinetics, however, $Fe^{2+}$ is unlikely to act as a direct abiotic U(IV) reductant. More likely are crystalline Fe oxides, which have shown to reduce U(VI) to U(IV) at near–neutral pH (Ginder-Vogel and Fendorf, 2008). At Gullmar Fjord, crystalline Fe oxides (F3) are observed from ~10 cm and onwards. They remain relatively constant throughout the sediment core, by which they could be involved in U(VI) reduction and U(IV) precipitation.

Our geochemical data reveals that U mostly resides in clay minerals (F6) or associated with refractory OM complexes (F4). However, U in F4 and F6 strongly covaries with Al in F4 and F6 (Fig. S6) and remain relatively constant throughout the sediment core. When U(VI) builds complexes with OM, it becomes unresponsive to microbially mediated U(VI) reduction – despite favorable reduction conditions – which decreases the pool of reactive and mobile U species (Ortiz-Bernad et al., 2004; Fuller et al., 2020). Therefore, we infer that U in F4 (OM-bound) and F6 (clay-bound) is of detrital origin and largely unreactive, rather than representing reactive authigenic U phases.

Instead, possible host phases of authigenic U are expected in F2, e.g., associated with Mn carbonates (F2), which particularly below 24 cm strongly covary with each other. Presumably, U extracted in F2 represents monomeric non-uraninite (Fu et al., 2018; Jokinen et al., 2020b). Water column monitoring phosphate ($PO_4^{3-}$) data indicate benthic release of $PO_4^{3-}$ during the sampling month (~5 µmol $L^{-1}$ at 110 m; SMHI, 2022), suggesting high pore water $PO_4^{3-}$ concentrations. Numerous studies have observed that $PO_4^{3-}$ inhibits the formation of crystalline uraninite (Bernier-Latmani et al., 2010, Boyanov et al., 2011; Alessi et al., 2014; Morin et al., 2016), supporting the presence of non-uraninite in F2 (Jokinen et al., 2020b). Remarkably, F2 and F3 follow a very similar downcore enrichment pattern (Fig. S7). In contrast to U in F2, U in F3 likely comprises crystalline uraninite, as inferred by laboratory experiments (Fu et al., 2018, uraninite extracted with hydroxylamine). To explain this co-variation and downward increase, it could be argued that U in F2 and F3 represent a continuum of U host phases extractable in both F2 and F3. Monomeric non-uraninite and crystalline uraninite are known to have a high affinity to form complexes with OM, by which OM could be a possible candidate as a common U host phase (Alessi et al., 2014; Bone et al., 2017). However, neither F4 nor $C_{org}$ shows a meaningful correlation with F2 or F3. Thus, association of U in F2 (non-uraninite) and F3 (crystalline uraninite) with OM is unlikely at this site (Figs. 4a, S7).

**Table 2.** Estimated authigenic Mo and U Mass accumulation rates ($TM_{MAR}$) of the upper 10 cm (whole core in brackets) and benthic fluxes ($TM_{benthic flux}$) at Gullmar Fjord (GF-117) and Koljö Fjord (KF-43). Positive benthic fluxes (+) refer to benthic release and negative fluxes (−) refer to benthic uptake. $TM_{MAR}$ were estimated using total solid-phase Al, Mo, and U concentrations derived from the sum of F1–F6 in the sequential extraction data. All values are given in µmol $m^{-2}$ $yr^{-1}$. Note that Mo- and $U_{benthic fluxes}$ provided here are the same as those provided in Fig. 2 given in nmol $m^2$ $d^{-1}$.

| Site | $Mo_{MAR}$ (UCC) | $Mo_{benthic flux}$ | $U_{MAR}$ (UCC) | $U_{benthic flux}$ |
|------|------|------|------|------|
| GF-117 | 87.15 (69.49) | +25.13 | 9.52 (20.56) | −2.04 |
| KF-43 | 238.91 (332.33) | −39.21 | 11.39 (17.98) | −2.12 |

**5.2.2 Molybdenum and Uranium in Koljö Fjord**

**Molybdenum**

In contrast to Gullmar Fjord, solid-phase Mo/U ratios at Koljö Fjord are elevated throughout the sediment core relative to average seawater, suggesting enhanced authigenic Mo, which is in line with elevated Mo-EFs above the crustal background and compared to U-EFs (Fig. 1c). Bottom water redox conditions are subject to inter–annual fluctuations, resulting in a range from oxic to sulfidic conditions (Fig. 1b). Such fluctuations favor formation of Fe- and Mn oxides, which are the most efficient scavenging and transport carriers of Mo in waters with variable oxic–suboxic boundaries (Scholz et al., 2013; Bertine and Turekian, 1973; Algeo and Lyons, 2006; Wagner et al., 2017). The six-fold higher Mo accumulation rates ($Mo_{MAR}$) of the upper 10 cm compared to the Mo benthic influx support this particulate shuttling

transport mechanisms for Mo at Koljö Fjord. In contrast to Gullmar Fjord, the sulfidic bottom and pore water permit

efficient Mo burial, and limit re-release to the bottom water (Sulu-Gambari et al., 2017; Lenstra et al., 2019).

In the sediment, the close relationship between Mo burial and Mn- and Fe oxide refluxing is demonstrated by the

corresponding fluctuations in Mo with Fe, and particularly, Mn enrichments (Fig. 4b). However, in sulfidic pore water,

Mn oxides are negligible as host phases due to reductive dissolution of such (Fig. 3b). Instead, our data suggest that Mo

weakly-bound to S-phases (F1, i.e., thiomolybdate intermediates) may serve as transitional host phases after release of

Mo from Mn oxides. The presence of these particle reactive thiomolybdate intermediates agrees with pore water $\Sigma H_2S$

concentrations >11 µM required subsequent replacement of O atoms with S atoms during subsequent thiolation

($MoO_4^{2-}$ to $MoS_4^{2-}$; Helz et al., 1996; Erickson and Helz, 2000), and explains the gradual decrease in pore water Mo/U

molar ratios.

Likely candidates for more permanent host phases of Mo are F3 and F4. These comprise crystalline Fe oxide

(F3), which in contrast to labile Mn oxides can survive sulfidic conditions (Hermans et al., 2021), as well as the labile

and more refractory OM pool (F3 and F4). Both fractions are of similar size and generally follow the same pattern as F1

suggesting that Mo initially stored in labile thiomolybdate intermediates on mineral surfaces (e.g., Helz et al., 1996;

Vorlicek et al., 2018) may also become more permanently incorporated into Fe oxides and OM (e.g., Chappaz et al.,

2014; Dahl et al., 2017). Organic-bound host phases for Mo are a common finding in different marine sediments

underlying suboxic–euxinic water columns (e.g., Huerta-Diaz and Morse, 1992; Algeo and Lyons, 2006; Scholz et al.,

2013). Other permanent Mo host phases frequently discussed are pyrite (e.g., Huerta-Diaz and Morse, 1992; Sundby et

al., 2004; Chappaz et al., 2014), and Fe-Mo-S colloids, such as $FeMoS_4$ (Vorlicek et al., 2018; Helz and Vorlicek, 2019;

Helz, 2021). Both mineral phases are expected to be extracted in F5 (Table 1). While adsorption of Mo onto pyrite

surfaces have been considered a possible Mo sequestration pathway (Huerta-Diaz and Morse, 1992), in practice,

correlations between Mo and Fe or total S contents are often very weak (Lyons et al., 2003; Algeo and Maynard, 2004;

Scholz et al., 2013). Consistent with these studies, at Koljö Fjord, neither the correlation between F5 Mo and Fe:S nor S

is significant ($R^2 = 0.02$, and $R^2 = 0.03$, respectively). Thus, we infer that pyrite is probably negligible as a Mo host

phase. Although Fe-Mo-S colloids are more likely to host Mo in F5, given the small fraction size relative to the sum of

645 all fractions (median ~8 wt. %), the total contribution of Mo sequestered as Fe-Mo-S colloids is of minor importance.

**Uranium**

In contrast to Mo, benthic U fluxes are very similar between Gullmar and Koljö Fjord, illustrating that diffusion – as the

key sequestration pathway of U – commences in a similar manner under either oxic–dysoxic or suboxic–sulfidic bottom

waters. Similarly, estimated $U_{MARs}$ of the upper 10 cm are only slightly elevated over those at Gullmar Fjord but also

about five times greater than the U diffusive flux (Table 2). This implies that U water column dynamics (particulate transport to the sediment surface) behave relatively similarly between the two fjords. Despite a strong pycnocline and dysoxic–suboxic conditions below ~15–17 m water depth, Koljö Fjord surface waters are usually well-oxygenated. Under such conditions PNU preservation is also expected to be limited here (Zheng et al., 2002b), which makes an additional supply of U by Fe oxides more likely.

As Koljö Fjord's pore water become sulfidic just below the SWI, $U_{diss}$ is removed from the pore water within the upper 5 cm (Fig. 3b). Within the same depth interval, a small fraction of U is (loosely)-sorbed U(VI) to mineral phases (F1, Fig. 4b). However, with onset of reducing conditions, F1 rapidly decreases and is replaced by more refractory U(IV) mineral phases extracted in F2–F4. The majority of authigenic U resides in F2, followed by F3 and F4 with similar fraction sizes (Fig. 4b). Remarkably, all three fractions follow a similar downward trend (Fig. S7). They also show a strong covariation with $C_{org,}$ particularly with terrestrial $C_{org}$ (Figs. 4b, S7). As refractory metal-OM complexes are expected to be extracted in F2–F4 (Table 1), these findings imply that, unlike at Gullmar Fjord, the majority of authigenic U at Koljö Fjord is bound to labile and refractory OM complexes or $C_{org}$-coated minerals (Bone et al., 2017). Although some crystalline Fe oxides are present at Koljö Fjord (section 5.1.2.) to serve as potential reduction agent of U(VI), the strong inverse correlation between pore water $\sum H_2S$ and $U_{diss}$ – a phenomenon observed in the Black Sea water column (Rolison et al., 2017) – suggest that U(VI) reduction is dominantly enzymatically mediated by sulfate reducing bacteria (Lovley et al., 1993; Fletcher et al., 2010).

Sulfate reduction also has implications on type of U(IV) mineral phases precipitating from solution, since $SO_4^{2-}$ favors non–uraninite formation (Fletcher et al., 2010; Fuller et al., 2020). In other words, even under sulfidic conditions, U(VI) associated with OM may only be partially reduced to crystalline uraninite U(IV), and therefore the latter may not be the dominant U phase despite favorable reduction conditions (Cumberland et al., 2018; Fuller et al., 2020). This infers that at Koljö Fjord OM-bound U likely consists of a mixture of reactive (prone to remobilization) monomeric non-uraninite U(IV), unreactive (less prone to remobilization) crystalline uraninite U(IV), and oxidized U(VI) complexes (Sharp et al., 2011; Alessi et al., 2012; Jokinen et al., 2020b). Consequently, the results suggest that besides sulfidic conditions, the presence of OM may increase the preservation potential of U in fjord sediments.

Despite the reduction potential of reduced S species (i.e., FeS or pyrite, Bargar et al., 2013; Cumberland et al., 2021) on U(VI), U does not form direct bonds with such species (Choppin and Jensen, 2006; Bone et al., 2017). As supported by the apparent decoupling between U in F5 (pyrite extractable) and Fe:S, Mo, Fe, and S in F5. Such decoupling suggests that U neither forms Fe-S precipitates nor absorbs directly onto other metal-Fe-S phases, such as pyrite or Mo-Fe-S colloids. In contrast to the decoupling of Mo and U in F5 (and all other fractions), both trace metals

show a very similar enrichment pattern in F1 (Fig. S8). As the Mo-U covariation pattern is absent at Gullmar Fjord and thiomolybdate intermediates are the most logical host phase for Mo in F1 at Koljö Fjord, this could suggest that U is sorbed to those thiomolybdates. However, the relatively consistent ~2 cm offset – particularly below 20 cm – between Mo and U in F1 makes a presence of a common host phase very unlikely (Fig. S8). Thus, the covariation pattern must have a different source. Bone et al. (2017) explained U-S correlations by associations between thiol-S species with OM to which U is absorbed. This indirect bonding of U to S species via organic coatings could indeed explain why U in F5 is decoupled from (Mo)-(Fe)-S species but present in F1.

All OM complexes are expected to be extracted in F4, and thus no OM associations are expected in F5. In accordance with Gullmar Fjord, we observe that F6 in U covaries with F6 in Al, suggesting that U is of detrital origin. Uranium(VI) may be transported to the basin sorbed to clay minerals, such as illite, which is the dominant clay mineral (grain size fraction < 2 μm) in Gullmar Fjord and Skagerrak (Hassellöv et al., 2001). Sorption of U(VI) to clay minerals have been demonstrated in laboratory experiments (Bachmaf and Merkel, 2011; Mei et al., 2022). Strikingly, U residing in F6 is believed to represent the most refractory U host phase – crystalline uraninite (Jokinen et al., 2020b; F5 in their protocol). Our data does not support this hypothesis. This may be partially explained by the additional extraction step we conducted to separate the pyrite fraction (F5) and silicate fraction (F6).

**5.3 Key authigenic Fe, Mn, Mo, and U sequestration mechanisms at Gullmar and Koljö Fjord**

In summary, our geochemical data suggest that sedimentary authigenic Mo and U sequestration in Gullmar and Koljö Fjord is strongly controlled by differences in composition and availability of pore water and solid-phase species, particularly $\Sigma H_2S$ and OM, and Fe and Mn oxide shuttling (Fig. 5). At Gullmar Fjord, sediment geochemistry is governed by particulate shuttling of Fe, and particularly of Mn. Here, Fe and Mn oxides represent the key carrier and sedimentary host phases for authigenic Mo. First, Mo is sequestered by Mn oxide in the surface sediments; with progressing diageneses the Mo host phase switches to poorly crystalline- and crystalline Fe oxides deeper in the sediment. While crystalline Fe oxides are involved in U(VI) reduction to U(IV), ultimately, U builds its own authigenic mineral phases either associated with $PO_4^{3-}$ or carbonate. A considerable fraction of U appears to be of detrital origin, which is largely unreactive and immobile.

At Koljö Fjord, sedimentary elemental dynamics are controlled by sulfate reduction and subsequent release of $\Sigma H_2S$ to the pore water. Given the sulfidic conditions, authigenic sequestration of Mo in Mn oxides is strongly limited; instead, Mo is buried in sulfide phases, such as thiomolybdates and Mo-Fe-S colloids. Authigenic U sequestration at Koljö Fjord is governed by a combination of pore water $\Sigma H_2S$ and high OM contents. Here, U largely resides with OM, either as OM complex or adsorbed to organic coatings of other mineral phases, including sulfides.

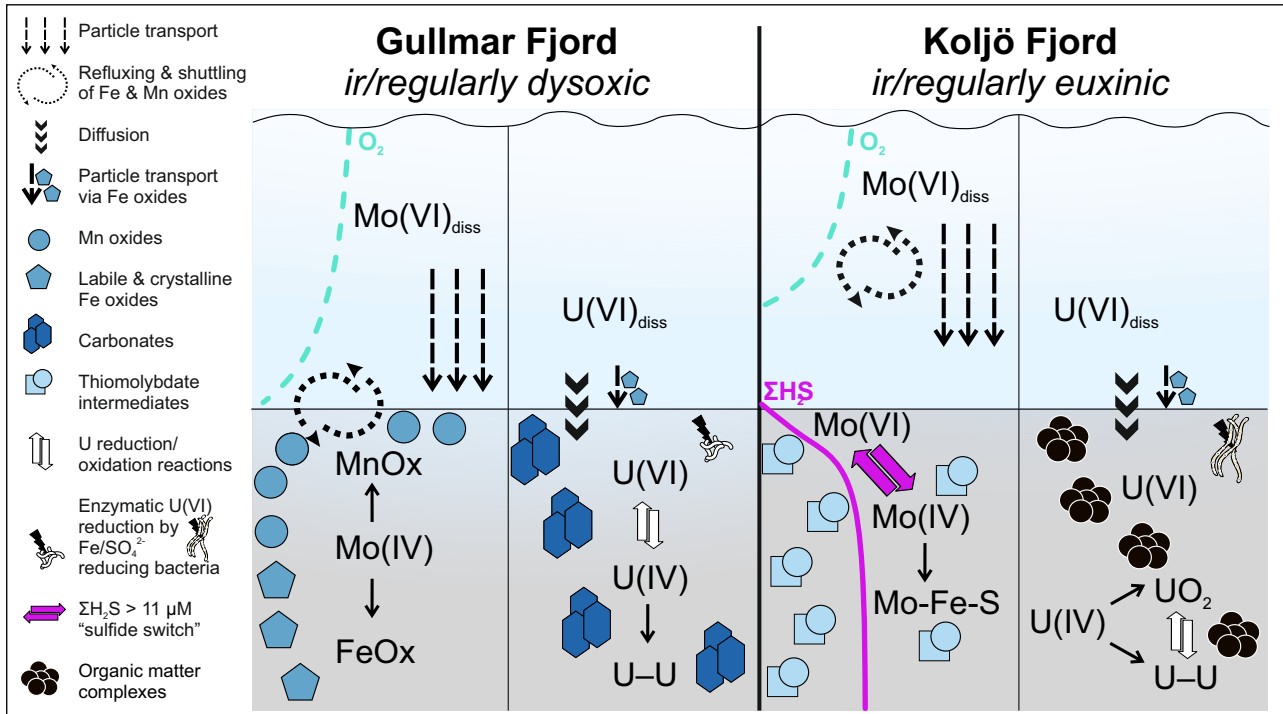

**Figure 5.** Summary of key authigenic Mo (left columns) and U (right columns) sequestration mechanisms at Gullmar Fjord (ir/regularly dysoxic) and Koljö Fjord (ir/regularly euxinic). $Mo(IV)_{diss}$ denotes the dissolved phase molybdate and $U(VI)_{diss}$ denotes dissolved uranyl complexes with e.g., carbonate or phosphate. All sedimentary Mo and U phases are present as solids. Within each panel, left to right indicates increasing content of solid-phases shown with symbols. For detailed description of sequestration mechanisms see section 5.2

**5.4 Applicability and constraints of Mo and U as paleo–environmental proxies in fjord environments**

Fjords are dynamic depositional systems characterized by high sedimentation rates, $C_{org}$ loading, and sensitivity to weather and climatic changes (e.g., Howe et al., 2010; Bianchi et al., 2020), which theoretically makes them desirable for paleo-environmental reconstructions. However, Mo and U redox proxy signals may be partially or entirely overprinted by secondary depositional environmental factors (Algeo and Lyons, 2006; Scholz et al., 2018; Jokinen et al., 2020b; Paul et al., 2023).

To verify whether authigenic Mo and U enrichments in fjord sediments can be reliably used as archives for changes in bottom water $O_2$, we assess to what extent distinct features in the trace metal speciation from Gullmar- and Koljö Fjord can be explained by hydrographic variability (e.g., occurrence of inflow events, or seasonal water mass exchange), or are a result of secondary controls pre-depositional (e.g., Fe and Mn oxide shuttling and water mass restriction), or post-depositional (e.g., oxidative remobilization and pore water geochemistry) factors.

**5.4.1 Molybdenum**

Sedimentary Mo enrichment patterns are frequently used to reconstruct temporal environmental changes in coastal marine settings related to de- or reoxygenation events, or (semi)-regular variability in water column stratification and

redox conditions, ranging from seasonal (e.g., Egger et al., 2016; Sulu-Gambari et al., 2017; Dellwig et al., 2021) up to centennial scale (e.g., van Helmond et al., 2018; Scholz et al., 2018). In non-euxinic settings, the efficiency of $Mo_{diss}$ removal and subsequent Mo sequestration into surface sediments is strongly coupled to ambient Mn redox dynamics, coupled to the redox variability in the water column and bottom water. In principle, short-term redox fluctuations in the bottom water promote Mn oxide refluxing, increasing the Mo flux to the sediment and subsequent surface sediment enrichment (Lenz et al., 2015b; Sulu-Gambari et al., 2017; Scholz et al., 2017; Dellwig et al., 2018).

Between 2010–2018, Gullmar Fjord's bottom water was largely dysoxic and only periodically interrupted by short-term oxygenation events (Fig. 2 and 4a). Likely combined with higher Mn oxide input into the fjord, these redox fluctuations promoted both Mn oxide (F3) and authigenic Mo accumulation (F3) in the subsurface (~upper 10 cm) sediments (Fig. 4a). As Mn oxides diagenetically convert into Mn carbonates over time (Burdige, 1993; Huckriede and Meischner, 1996), the presence of two additional Mn carbonates peaks (F2) at ~1990–2000 (~17–24 cm) and ~1960–1970 (~35–40 cm) suggest that there have been more than one period of enhanced Mn oxide refluxing in Gullmar Fjord, likely associated with oxygenated periods during gradually basin-wide deoxygenation after the 1960s (Fig. 2). One would expect that this trend towards more dysoxic conditions and enhanced Mn oxide shuttling have resulted in enhanced authigenic Mo enrichments in the surface sediment at that time. However, no record of such is visible throughout the sediment core, possibly due to poor Mo preservation potentials under the ambient low $\Sigma H_2S$ pore water concentrations at Gullmar Fjord.

To permanently preserve fluctuations in Mo sequestration, associated with e.g., seasonal changes in water mass stratification it requires both high Mo supply by Mn oxide shuttling and pore water $\Sigma H_2S$ concentrations > 11 µM (Egger et al., 2016; Sulu-Gambari et al., 2017) to lock Mo in sediment e.g., by the formation of stable Mo sulfides (Helz et al., 1996). Manganese oxide supply into Gullmar Fjord is naturally exceptionally high compared to other coastal marine systems worldwide (Burdige, 1993; Aller, 1994; Jokinen et al., 2020b; Lenstra et al., 2021b), and due to gravitational focusing of these Mn oxides at the deeper fjord (our study site; Brinkmann et al., 2023b), Mo supply is expected to have been sufficiently high throughout the past 80 years. Instead, upon reductive dissolution of Mn oxides under suboxic conditions in the surface sediments, insufficiently high pore water $\Sigma H_2S$ fuels benthic release of $Mo_{diss}$, and thereby reducing sedimentary Mo sequestration (Fig. 3a and 4a, Goldberg et al., 2012). Therefore, neither the initial signs of deoxygenation, nor the temporal variability in shuttling are recorded by Mo, which impedes the applicability of Mo as a redox proxy to reconstruct environmental changes at Gullmar Fjord.

In contrast to Gullmar Fjord, pore waters at Koljö Fjord are sufficiently sulfidic to permit permanent sequestration of Mo (Fig. 4b). Five distinct enrichment peaks in Mo were found at Koljö Fjord, all of which cover a

period of at least ten years: ~1860–1870, ~1923–1940, ~1944–1962, ~1972–1994, and ~2005–2013 (Fig. 4b). This eliminates the possibility that those Mo enrichment peaks are coupled to seasonal or episodical short-term redox fluctuations, either linked to rapid reoxygenation events as described for the Baltic Sea (Scholz et al., 2018; Dellwig et al., 2018), or seasonal variability in water column stratification and redox conditions, as described for a former estuary with comparable seasonal redox dynamics as at Koljö Fjord (Egger et al., 2016).

Instead, it is more likely that Mo enrichment peaks represent longer-term environmental changes. Although water column data show a trend towards more reducing bottom water conditions over the last century, the three Mo peaks for which monitoring data are available (~1944–1962, ~1972–1994, and ~2005–2013) seem to have been deposited under different environmental conditions. Between 1944–1962, the Mo peak was formed during non-euxinic conditions, while between 1972–1994 and 2005–2013 bottom waters were largely suboxic–euxinic, interrupted by oxygenation events of variable intensity and duration (Fig. 2). Therefore, Mo sequestration has in the last century migrated from the pore water– which must have been sulfidic shortly below the sediment-water interface to prevent benthic escape of $Mo_{diss}$ (Erickson and Helz, 2000) – to the bottom water (and water column). Seemingly, both mechanisms result in similarly strong Mo sequestration and preservation at Koljö Fjord, which is in line with high Mo contents in coastal sediments with a shallow SMTZ, underlying an oxygenated water column (Jokinen et al., 2020b). Such substantial effects of pore water chemistry on Mo sequestration complicate the application of Mo as a single proxy to reconstruct distinct environmental changes in this (and similar) fjord system(s).

**5.4.2 Uranium**

Uranium is considered as a more sensitive recorder of mild deoxygenation compared to Mo due to a higher reduction–oxidation potential of U relative to Mo (Lovley et al., 1991; Zheng et al., 2002a; van Helmond et al., 2018). According to this, the U record at Gullmar Fjord shows more variability than Mo, specifically in the reactive U(IV) monomeric non-uraninite pool associated with (Mn) carbonates (F2). Given the strong dependency of Mn oxide and -carbonate accumulation/ preservation on redox changes, U in F2 may provide some information about environmental changes. In fact, U in F2 shows more variability than Mn carbonates, implying that U might have recorded more redox changes. However, individual U peaks cover at least decade long timescales; as the redox changes occurred on shorter (seasonal) timescales, U peaks cannot be linked to individual/seasonal oxygenation events (Fig. 2 and 4a). Either U did not record these short-term redox changes, or the original redox signal has been overprinted by post-depositional redox-induced remobilization (e.g., Wang et al., 2013).

At Koljö Fjord, the timing of U minima and maxima is comparable to that of Mo, although the onset of enrichment peaks seems to occur slightly earlier in U (Fig. S8). This suggests that U might have recorded the timing of

redox variability more reliably than Mo and which – in addition to Gullmar Fjord – would support the idea of U being a more sensitive redox recorder (e.g., van Helmond et al., 2018). Overall, the three broader U peaks occurring between ~1920 and ~1980 (Fig. 4b) coincide with longer periods of oxic–dysoxic intervals intercepted by somewhat periodic suboxic–euxinic intervals (Fig. 2 and 4b). During these intervals, U in F1 shows a strong covariation with Fe in F2 (Fig. S9a) and Fe and Mn in F3 (Fig. S9b), suggesting a stronger particulate flux of U by Fe (and Mn) oxides. In contrast to Mo, the (for Koljö Fjord) unusual long oxic period between 1993 and 1997 is detectable by a clear U minimum (F1) and coinciding small Mn and Fe oxides (F3) enrichment peaks (Fig. S9b), highlighting the better preservation conditions for labile Mn and Fe oxides. Sequestration of loosely sorbed (labile) U was probably limited by higher bottom water $O_2$, which reduced the reduction potential into more refractory phases. The most recent U peaks (~2007 and ~2015–2017, Fig. 4b) are narrower and more distinct compared to the older ones. In correlation with the pronounced Fe in F2 peaks and monitoring data (Fig. 2), the preservation of these U peaks may be linked to two environmental changes: (1) rapidly occurring individual oxygenation events following or followed by euxinic conditions; or in case of the most recent U peak, (2) a sign of more consistently euxinic conditions, since the rise in U (after ~2012) coincides with the longest euxinic period recorded at Koljö Fjord (Fig. 2). In contrast to Gullmar Fjord, U provides some degree of paleo redox proxy potential at Koljö Fjord for the recent past, but distinct changes dating back more than ~15 years ago cannot be identified. Even under these more reducing conditions, the original U redox signal may be obscured by post-depositional remobilization. In absence of $O_2$, other dissolved or solid-phases such as nitrate, Fe(III) oxides, FeS, or dissolved carbonate can directly or indirectly oxidize U(IV) (e.g., Ginder-Vogel et al., 2006; Alessi et al., 2012; Wang et al., 2013; Bi and Hayes, 2014).

**5.5 Implications for using Mo and U-based environmental (redox) proxies in fjords sediments**

Our findings suggest that using Mo and U enrichments in fjord sediments as archives of environmental changes hosts certain pitfalls. While environmental changes over decadal time scales may be identifiable, higher frequency events (seasonal or episodical) are either not reliably recorded by Mo and U, due to non-steady-state depositional conditions, or overprinted by post-depositional processes. Thus, neither at Gullmar Fjord nor at Koljö Fjord, it is possible to accurately reconstruct environmental (redox) changes on these timescales using Mo and U enrichments. This is in-line with observations made at the Canadian fjord, Saanich Inlet (Algeo and Lyons, 2006), where the occurrence of high frequency events partially restricts the applicability of the Mo/TOC ratio – a widely applied measure for the degree of basin restriction (Algeo and Lyons, 2006). Unsurprisingly, in our sedimentary records from Gullmar and Koljö Fjord, significant correlations between TOC and Mo are lacking as well (Fig. S10). Therefore, when describing the degree of

water mass restriction in fjord settings, is not advisable to solely consider the Mo/TOC ratio, as this approach may result in false interpretations.

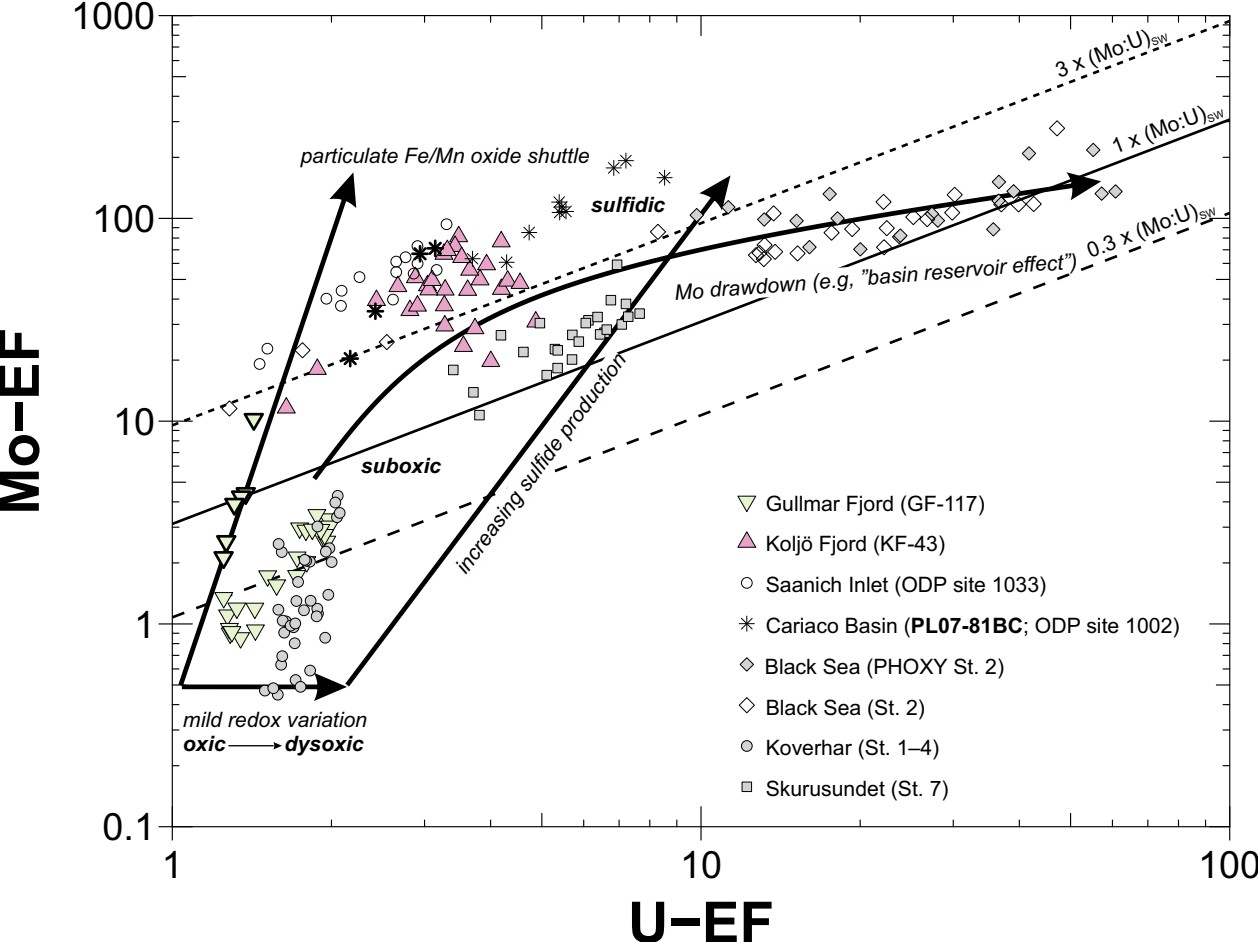

**Figure 6.** Sedimentary Mo-EF and U-EF covariation patterns at Gullmar Fjord (light green-filled reversed triangles) and Koljö Fjord (pink-filled triangles), relative to other coastal marine environments worldwide (unfilled symbols). Grey filled symbols depict data from a parallel study, Koverhar (St.1–4, circles), Skurusundet (St. 7, squares) and Black Sea (PHOXY St. 2, diamonds) (Paul et al., 2023). The four black arrows and letters denote key enrichment controls as described in Algeo and Tribovillard (2009) and the three diagonal lines denote multiples (0.3, 1, and 3) of the present-day seawater (SW) Mo:U ratio converted to an average weight ratio of 3.1 for the purpose of comparison with sediment Mo:U weight ratios (Tribovillard et al., 2012). Additional literature data are from Yano et al. (2020) – Saanich Inlet (ODP site 1033, unfilled circles), Cariaco Basin (ODP site 1002, plain stars), and Black Sea (St. 2, unfilled diamonds) – and Calvert et al. (2015) – Cariaco Basin (PL07-81BC, four samples, bold stars). Gullmar Fjord data is divided into two sub-patterns, the upper sediment section (0–7 cm) with the steepest Mo-/U-EF ratios, and the remaining sediment core (7–59 cm) with less steep Mo-/U-EF ratios (see text).

Despite these limitations, sedimentary Mo and U enrichments may still be applicable as environmental (redox) proxies when considering total Mo and U (enrichment factors), expressed as ranges over the whole sediment core (Fig. 1c and d) or as Mo- and U-EF cross-plots, (Fig. 6, Algeo and Tribovillard, 2009). In fact, sedimentary Mo- and U-EFs covariation patterns at Gullmar- and Koljö Fjord (Fig. 6) clearly support the key sequestration mechanisms inferred by our sequential extraction data (Fig. 4). Both sites follow the Fe and Mn oxide shuttling signature, although it is more distinct at Gullmar Fjord, given the more intense Fe and Mn oxide cycling. This is reflected by a clear division into

higher Mo-/U-EF ratios, associated with the shuttling-induced surface maxima of Mo (0–7 cm, bold reversed triangles, Fig. 6), and lower Mo-/U-EF ratios in the remaining core (> 7 cm).

The central role of $\sum H_2S$ in controlling Mo sequestration and burial efficiency is demonstrated by the clear separation of Gullmar Fjord and Koljö Fjord by the 1x molar Mo:U seawater line, dividing non-sulfidic (bottom water) sites (Koverhar and Gullmar Fjord) from ir/regularly sulfidic sites (Saanich Inlet, Cariaco Basin, Skurusundet, Black Sea, and Koljö Fjord). This distinct Mo- and U-EF pattern corresponds well to our geochemical data revealing that despite strong Mo shuttling by Fe/Mn oxides (i.e., at Gullmar Fjord), ultimately, inadequate bottom- and pore water chemistry (here: low $\Sigma H_2S$) is the key factor explaining lower Mo-EFs compared to other ir/regularly dysoxic sites (e.g., Lilla Värtan, Stockholm Archipelago), by limiting permanent Mo sequestration. Correspondingly, total U enrichments are highly sensitive to the presence of dissolved and solid-phase OM, carbonate, and phosphate phases. All these phases can result in precipitation of refractory and inert U mineral phases, which can help to explain both lower U-EFs under more reducing conditions, and higher U-EFs under less reducing conditions. Evidently, pore water geochemistry (i.e., composition and concentration of dissolved phases) dictates permanent Mo and U sequestration in our fjord sediments, regardless of the bottom water redox condition and the impact of secondary pre-depositional factors (i.e., Fe- and Mn oxide shuttling).

**Conclusions and outlook**

Our trace metal sequestration extraction-based case study highlights how environmental and post-depositional factors may obscure direct information about past redox conditions stored in sedimentary Mo and U enrichments in fjord settings. We further complement the current understanding of Mo and U sequestration in fjord sediments, albeit considerable limitations regarding U speciation. This demonstrates the urge for future studies, ideally combining both 1) a modified sequential extraction scheme with targeted U extraction steps to differentiate between crystalline uraninite and monomeric non-uraninite species, and 2) microanalytical techniques, such as synchrotron-based X-ray absorption spectroscopy (e.g., XANES, EXAFS) or Nano Secondary Ionizing Mass Spectrometry (nanoSIMS), providing more information on metal speciation and host phases (e.g., oxidation states, coordination number, and isotopic composition). Despite these limitations, the key findings from this study are:

- *Mo* enrichment factors (EFs) are *applicable* as a comparative redox proxy for whole sediment core data in fjord settings, when considering a *certain* degree of diagenetic *overprinting* of initial redox signals.

- *U* enrichment factors (EFs) are *applicable with caution* as a comparative redox proxy for whole sediment core data in fjord settings, due to *considerable* diagenetic *overprinting* of initial redox signals.

- Permanent *U sequestration is more complex than Mo*; even in less reducing sediments more refractory mineral phases may be formed and vice versa strongly reducing sediment may not guarantee stronger U enrichments.

- *Temporal variability in Mo and U* enrichments can be used to detect environmental changes over *decadal time scales*, whereas *higher frequency* events (seasonal or episodical) are likely *not* being *recorded* or are *overprinted* by post-depositional processes.

- In *dynamic fjord-type settings*, it is *advised to restrain* from reconstructing environmental changes based on individual Mo and U enrichment profiles, however existing environmental (redox) proxies, such as *Mo- and U-EF covariation patterns*, may still retain reliable *informative value.*

**Data availability**

Research data are available from Zenodo (10.5281/zenodo.8399270) or upon request.

**Competing interests**

The authors declare that they have no conflict of interest.

**Author contributions**

KMP, MH, and TJ designed the study. SAJ, IB, and HLF organized the fieldwork and carried out the sampling. KMP, SAJ, and TJ conducted the pore water analyses. KMP and MH executed the sequential extractions. KMP, MH, and TJ interpreted the data and wrote the paper, with comments provided by SAJ, IB, and HLF.

**Acknowledgements**

We thank the captain, crew, and scientists on board R/V Skagerrak in September 2018. We acknowledge the staff of the Kristineberg Marine Research Station. The HelLab technicians, especially Juhani K. Virkanen and Tuija B. Vaahtojärvi are acknowledged for their analytical assistance at the Department of Geosciences and Geography, University of Helsinki. We further thank Heini Ali-Kovero for the C/N analyses at the Ecosystems and Environment Research Programme, University of Helsinki.

**Funding**

This research was funded by the Academy of Finland (grant no. 1319956 and 1345962), and the Onni Talas Foundation. HLF acknowledges support by the Swedish Research Council VR (grant no. 2017-04190).

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
