# Peer review of "Revisiting the applicability and constraints of molybdenum and uranium-based paleo redox proxies: comparing two contrasting sill fjords"

_Biogeosciences, 2023_

## Author Response (AR1)

**Author's response**

Dear Dr. Crowe,

We thank you and the two anonymous referees for their constructive comments and suggestions to improve our manuscript bg-2023-83 "Revisiting the applicability and constraints of molybdenum and uranium-based paleo redox proxies: comparing two contrasting sill fjords".

We have considered all comments and suggestions and implemented them accordingly. In our line-by-line responses to both referees, we address all modifications made to the text and figures in the manuscript and supplementary material. The line numbers refer to the revised manuscript, unless indicated otherwise. A record of all modifications can be obtained from the "tracked changes" version of the manuscript.

In addition to these requested changes, we have also made minor modifications and corrections to the text and figures. As of 05.10.2023, our research data are publicly accessible from the data repository Zenodo (10.5281/zenodo.8399270).

Kind regards,

on behalf of all co-authors

Mareike Paul
* * *
**Anonymous Referee 1:**

*Summary*

Paul et al present a study of two silled fjords with varying bottom-water redox conditions, assessing the impact of redox and other parameters on trace metal enrichment (namely Mo and U) in sediments. They use water column sensor data for an O2/H2S time series, and conducted several types of analyses on one sediment core recovered from each fjord in 2018. Specifically, they analyzed porewater elemental concentrations, C/N ratios of organic matter, and major + trace element concentrations in operationally-defined solid phase fractions from a sequential leaching method.

This is a wealth of data about a topic of interest to (paleo-)oceanographers: trace metal enrichment in reducing sediments. The contrasting water mass chemistry and basin hydrography of these fjords makes for a nice setting to test the effects of various processes. While I find the analytical effort quite commendable, the sequential leaching data have only a limited ability to provide definitive answers to many questions of interest, since multiple potential Mo and U host phases could be dissolved in any given step. Without further constraints on Mo and U delivery or exact host phases (e.g., from XANES/XAFS, Raman spectroscopy, or further-developed sequential leaching), we are left wondering exactly what processes govern ultimate Mo and U burial at each site. Despite this inherent limitation, though, these data add to the growing literature on this topic, and will certainly be useful for the field to build upon. I therefore recommend that this paper ultimately be published following some revisions to clarify the analyses/calculations undertaken, the calculations they inform, and their relationship to similar work at other sites around the world.

*Over-arching comments*

I have two over-arching comments for the authors to consider.

First, I think these results could be better situated in the context of parallel work in other restricted anoxic basins worldwide. Many sites with similar basin hydrography and redox have been studied for sedimentary Mo and U dynamics. The Black Sea is the best studied of these, but similar work exists from Saanich Inlet, Framvaren Fjord, Cariaco Basin, Santa Barbara Basin, etc. While processes of interest are discussed here with citation to work on other basins, a direct comparison of these data (e.g., Mo and U concentrations, Mo/TOC and U/TOC, Mo/U enrichments) to the complementary existing data from other basins in the literature would considerably strengthen this paper and make it more broadly relevant to the community. It would seemingly be easy to make some plots that include data from other basins, which would help the reader grasp what is similar/different in these fjords.

Second, and building on the point above, I think the authors should lead with a more thorough consideration of the bulk-sediment geochemistry using parameters that are widely reported for similar basins worldwide (i.e., Mo/TOC, U/TOC, Mo/U). The phase-specific digestions are a nice way to unpack additional detail, but would be better framed in the context of the bulk trace metal geochemistry. This would allow an initial discussion of net budgets of Mo and U in these basins, their sources/sinks and residence time, and comparison to dynamics in similar basins worldwide. Then the phase-specific data could be used to try to test further hypotheses, and ultimately culminate with the discussion about the use of Mo and U as short-timescale (~decadal) redox proxies (which I agree, is complicated due to ongoing redox perturbations in porewaters). Some of this discussion already exists in the supplement currently, but would be best included in the main text, along with figures to highlight these comparisons.

**Reply: We support the suggestion to directly compare our fjord data to other previously investigated sites. We have followed this suggestion by adding two figures, Mo-EF/U-EF (Fig. 6) and Mo/TOC (Fig. S10 – in the supplement), which are discussed in section 5.5 The selection of study sites was based on the comparability of the depositional environment, considering similar sites, i.e., Saanich Inlet (François, 1987); Framvaren Fjord (Skei et al., 1988 from Algeo and Lyons, 2006), Cariaco Basin (Lyons et al., 2003; Calvert et al., 2015; Yano et al., 2020), Koverhar (Jokinen et al., 2020b; Paul et al., 2023), and Skurusundet (Paul et al., 2023). This selection was further limited by the availability of modern sedimentary data, which differed for Mo, U, and TOC. Generally, U data were less consistently available, i.e., missing entirely for modern sediments at Saanich Inlet and Framvaren Fjord.**

*Line-by-line comments*

Line 219: Were the blank corrections analytical or procedural? In other words, did blank contribution from the sequential extraction step get subtracted? If so, how large was this blank relative to the sample signal?

**Reply: Blank corrections were made both analytically and procedurally. For reference, we have included a table in the supplementary material (Table S1a, b) that shows all the blank values for U, Mo, Mn, Fe, Al, Ca, and S for each fraction.**

Line 229: Were totals as determined by summed fractions compared to totals measured on samples simply subjected directly to the most intense bulk-digestion protocol?

**Reply: We compared totals determined by summing fractions and total digestion (available from Paul et al., 2023) during data analyses. Procedural and analytical differences between the two methods (e.g., sample inhomogeneity and different ICP-MS instruments), resulted in varying offsets between the total digestion (TD) and sequential extraction (SE), as shown here for Mo and U data.**

|  | Gullmar Fjord | | Koljö Fjord | |
| --- | --- | --- | --- | --- |
|  | Mo | U | Mo | U |
| Absolute difference between TD and SE (median, ppm) | 0.3 | 1.0 | 7.1 | 0.6 |
| Percentage difference between TD and SE (median, %) | 13.6 | 27.4 | 18.5 | 6.4 |

**While there is generally good agreement between the total digestion and sequential extraction data, both in the ranges and shape of the profiles, we consider our approach to use summed totals of fractions 1–6 as most valid when discussing total contents of each element.**

Line 248: These C/N ratios – particularly the terrestrial biomass value – are quite variable in reality. While this doesn't mean this sort of calculation is useless, I think it would be most appropriate to report an uncertainty on these estimations by using a plausible range of C/N ratios for both terrestrial and marine biomass, rather than just the preferred values cited here.

**Reply: We acknowledge this and have added a statement in the methods to indicate the ranges of end-member values suggested by Goñi et al. (2003). We also report the absolute ranges depending on the choice of the endmembers, both in Fig. 4 and in tabular form in the research dataset, which are available from Zenodo (10.5281/zenodo.8399270) from 05.10.2023.**

Line 269: The assumption that the lowest TM/Al ratios resemble the composition of incoming detrital material has significant implications for the results. Given that TM accumulation rates are calculated using the calculated TMXS, the fact that lower-than-average-crust TM/Al ratios are used for the baseline makes the difference of inferring that these sediments are a sink rather than source of trace metals to seawater. I see that this approach was taken in two earlier cited studies, but in those I also don't see further justification for this assumption, rather than simply citing that negative enrichments are obtained if using the global average upper continental crust TM/Al ratios. I think further substantiating this claim is important for demonstrating the legitimacy of the calculated TM enrichment rates. One way to do this would be via compilation of TM/Al ratios in surrounding lithology. Another would be to compare to other sediment samples in each fjord as a function of depth and distance from shore. Yet another would be to look at coupled bottom water and porewater concentration profiles.

**Reply: This is a valid and important remark. We do not have such data available, so for simplicity, we have now chosen to use the UCC value for the estimates, as described in section 2.4.4. One consequence of this is that for a very low number of samples (n=5) TM$_{XS}$ are negative (specifically Mo$_{XS}$ at Gullmar Fjord). These values are omitted from the estimation of accumulation rate of authigenic TM.**

Line 323: U negatively covaries with H2S in Koljo Fjord porewaters, consistent with what is seen in the deep waters of the Black Sea (e.g., Rolison et al 2017 GCA).

**Reply: We thank the reviewer for this important observation. We highlight this correlation now in the discussion section 5.2.2 (lines 663–666).**

Line 335: As noted above, the C/N ratio assessment of terrigenous versus authigenic marine biomass input is plagued by uncertainty in the composition of each end member. Here when discussing quantitative inferences of end member organic input using sediment core data, the calculation is further plagued by potential diagenetic alteration of the C/N ratio in the water column or sediments during remineralization. In fact, anaerobic remineralization can elevate the C/N ratio (e.g., Van Mooy et al 2002 GCA). So given the low-O2 nature of both sites, the measured C/N ratios of ~11 could derive from greater marine plankton input and subsequent alteration. This is not something we can precisely know, so I bring this up to simply acknowledge the uncertainty in this estimation of organic matter sources.

**Reply: We acknowledge the reviewer's comment and address the possibility of diagenetic overprint in section 4.2. (lines 351-353).**

Fig. 2, Lines 530-578: I'm still wondering where exactly most U resides in each case. The silicate fraction is larger in Gullmar, consistent with smaller U enrichments given less strongly reducing conditions (i.e., no H2S). However, if carbonate-hosted U is dominating the F2 signal, why is there less U in F2 in Gullmar – where Ca is abundant – than in Koljo?

Related to this is the question of which phase would hold any uraninite-hosted U, which could be the product of local U reduction (though the point is made that perhaps non-UO2 is the reduced phase here is sulfate-reducing bacteria are responsible for most U reduction, which is a valid inference). Another layer of uncertainty is that U can be (/likely is) complexed to organics, which might come out in F2, F3 or F4.

**Reply: These are valid points made by the reviewer. In general, with our available data, we can only hypothesize on the most likely host phases and on the exact speciation of U (non-uraninite and crystalline uraninite). This is partially related to the fact that with our extraction scheme more than one phase is extracted in F2. In addition to the mentioned carbonates, also FeS, Mn(II) phosphates, labile Fe oxides, and labile OM complexes, are being extracted.**

**As outlined in sections 5.2.1 and 5.2.1 (uranium), based on our pore-water and solid phase data, we believe that U host phases extracted in F2 in fact differ between the two fjords (with reference to Figs., S6 and S7.**

**For example, in Fig. S7, we can clearly see that at Koljö Fjord U in F2 (and F3 and F4) covaries with Corg, arguing for OM-bound U phases extracted in F2. At Gullmar Fjord, we cannot see such covariation of F2 (and F3) to Corg and F4 (where we expect to extract most refractory OM), implying that U in F2 is not bound to OM complexes.**

**Therefore, lower U in F2 contents at Gullmar Fjord compared to Koljö Fjord is likely due to the different host phases extracted in F2 combined with the contrasting behavior of U at each site.**

**Based on Fu et al. (2018), who extracted crystalline uraninite with hydroxylamine, we suspect that at Gullmar Fjord crystalline uraninite is hosted in F3.**

**We made a few minor clarifications to section 5.2.1. (lines 585-604).**

Section 5.2: Here it would be helpful to use plots such as those in Algeo & Lyons (2006) and Algeo & Tribovillard (2009) (which are already in the reference list). Specifically, a plot with Mo and U vs. TOC would help elucidate the impact of TOC on Mo and U enrichment, as well as the impact of basin restriction on metal enrichments (since you could also include data from other anoxic basins, such as those in the Algeo papers). Similarly, Mo (EF) vs U (EF) plots would help to visualize Mo vs U dynamics in each fjord. Even if the "reference" value for calculating the EF is different here, it is the relative changes that will be of interest, and potentially reflect basinal redox dynamics.

**Reply: We followed the reviewer's recommendation. For a detailed response, we refer to our response to the overarching comment by reviewer #1.**

Line 792: In order to confirm that "inadequate pore water chemistry" is the reason for lower Mo (EF) here than in other sites with similar bottom water redox, it would be useful to compare bottom water [Mo] across those settings (& same goes for the U discussion). As is seen clearly in Algeo & Lyons (2006), Mo/TOC correlates strongly with [Mo] in anoxic silled basins (their Fig. 8a).

**Reply: We agree with the reviewer that such a comparison would strengthen our hypothesis. However, as seen from the lack of any meaningful Mo-TOC correlations at both sites (due to high frequency events, as described for Saanich Inlet), no meaningful relationship to the conceptual**

**model of Algeo and Lyons (2006) can be established. For comparison, we added a Mo/TOC plot (Fig. S10) to the supplementary material.**

Line 800: As noted above, these Conclusions would be strengthened by comparing these data to other basins where similar work has been conducted. For instance, deeming U burial more complex than Mo is contingent on redox conditions, etc.

**Reply: We followed the reviewer's recommendation. For a detailed response, we refer to our response to the overarching comment by reviewer #1.**
* * *
**Anonymous Referee 2:**

General comments:

This manuscript aims to compare Mo and U paleoredox proxies under contrasting depositional settings in two fjords (i.e. low oxygen and euxinic). This approach is sound and provides much needed additional data on Mo and U geochemistry in modern sediments. The results appear to be of high quality and are presented in high quality figures and tables.

Specific Comments:

The overall discussion of the geochemical data is excellent – it is comprehensive and well-referenced. However, as discussed by Referee #1 in their review, much of the bulk geochemical data is not presented or discussed in the manuscript. I agree with Referee #1 that this bulk geochemical data should be included in the manuscript, prior to presenting the results of the sequential extractions. Comparison of this bulk data with data from other sedimentary environments would assist with placing these results into a wider, more useful, context (much of this data has been synthesised in recent reviews, so this should not be an onerous task). This is particularly important given that the studied fjords differ substantially in their sedimentation rates (i.e. Koljo Fjord's sedimentation rate is about half that of Gullmar Fjord).

**Reply: We refer to the response to reviewer #1**

Sequential extraction procedures are certainly useful for assessing possible host phases/speciation of trace elements in sediments, and the authors should be commended for accurately reporting their sequential extraction data without over-interpretation (which is unfortunately all too common). I am concerned that the sequential extractions were done on freeze dried sediments, rather than fresh sediments – work from Rapin et al. (1986) (amongst others) has shown freeze drying of sediments prior to sequential extraction can result in substantial changes to metal speciation compared to fresh, wet sediments (they recommend frozen storage as wet sediment to minimise disturbance of metal speciation). The authors should justify their choice of freeze drying as a preservation technique and discuss the possible implications of this on their results.

**Reply: We agree with the reviewer that performing sequential extraction on freeze-dried samples may alter the metal speciation as demonstrated by various studies. However, we carefully assessed different extraction methods and concluded that using freeze-dried samples was for our study logistically and methodically the most suitable method. There are very good reasons to use freeze-dried sediments, including the difficulty to homogenize wet samples and to determine the actual weight of the solids. Exposure to oxygen before and after freeze-drying was minimized by directly transferring the samples to an inert atmosphere.**

**Our approach is supported by numerous previous works using dried samples for sequential extraction, e.g., Poulton and Canfield (2005), März et al., 2008; Kraal et al., 2012; Jokinen et al., 2020a,b; Lenstra et al., 2021). Jokinen et al. (2020a,b) performed an almost identical procedure**

**to the one using in our study, and noted that there was no evidence for remobilization of highly redox-sensitive elements such as As from sulfide to oxide-bound fractions, as reported by e.g. Huang et al. (2015).**

**We have now added information on our steps in reducing oxygen exposure during sample treatment and clarified our motivation in using freeze-dried samples section 2.4.1 .**

It would be valuable to include some reflection on how the limitations of sequential extractions could be addressed in future studies to further refine our understanding of Mo and U behaviour in these fjords. Perhaps an assessment of the viability of using Synchrotron-based X-ray spectroscopy (e.g. XANES, EXAFS) to provide additional information on Mo and U speciation. These techniques are becoming increasingly accessible, but there is a lack of research investigating the reliability of sequential extraction procedures by comparison with Synchrotron-based speciation analysis of the same samples.

**Reply: We followed the reviewer's recommendation by acknowledging the limitations and recommendations for future research in the conclusions and outlook section.**

Figure 5 is an excellent inclusion in the manuscript to synthesise a rather complex discussion – well done!

**Reply: We greatly appreciate and thank the reviewer for their compliment.**

Technical corrections:

L384 – separated, not separated

**Reply: Corrected**

L666 – should be µm not µg

**Reply: Corrected**

**References**

Algeo, T. J. and Lyons, T. W.: Mo-total organic carbon covariation in modern anoxic marine environments: Implications for analysis of paleoredox and paleohydrographic conditions, Paleoceanography, 21, doi: 10.1029/2004pa001112, 2006.

Calvert, S. E., Piper, D. Z., Thunell, R. C., and Astor, Y.: Elemental settling and burial fluxes in the Cariaco Basin, Marine Chemistry, 177, 607-629, doi: 10.1016/j.marchem.2015.10.001, 2015.

François, R.: Some aspects of the geochemistry of sulphur and iodine in marine humic substances and transition metal enrichment in anoxic sediments, PhD dissertation, University of British Columbia, Vancouver, B.C., Canada, 462 pp., doi: 10.14288/1.0053223, 1987.

Fu, H. Y., Zhang, H., Sui, Y., Hu, N., Ding, D. X., Ye, Y. J., Li, G. Y., Wang, Y. D., and Dai, Z. R.: Transformation of uranium species in soil during redox oscillations, Chemosphere, 208, 846-853, doi: 10.1016/j.chemosphere.2018.06.059, 2018.

Goñi, M. A., Teixeira, M. J., and Perkey, D. W.: Sources and distribution of organic matter in a river-dominated estuary (Winyah Bay, SC, USA), Estuar Coast Shelf S, 57, 1023-1048, doi: 10.1016/S0272-7714(03)00008-8, 2003.

Huang, G. X., Chen, Z. Y., Sun, J. C., Liu, F., Wang, J., and Zhang, Y.: Effect of sample pretreatment on the fractionation of arsenic in anoxic soils, Environ Sci Pollut R, 22, 8367-8374, doi: 10.1007/s11356-014-3958-5, 2015.

Jokinen, S. A., Jilbert, T., Tiihonen-Filppula, R., and Koho, K.: Terrestrial organic matter input drives sedimentary trace metal sequestration in a human-impacted boreal estuary, Sci Total Environ, 717, 137047, doi: 10.1016/j.scitotenv.2020.137047, 2020a.

Jokinen, S. A., Koho, K., Virtasalo, J. J., and Jilbert, T.: Depth and intensity of the sulfate-methane transition zone control sedimentary molybdenum and uranium sequestration in a eutrophic low-salinity setting, Applied Geochemistry, 122, doi: 10.1016/j.apgeochem.2020.104767, 2020b.

Kraal, P., Slomp, C. P., Reed, D. C., Reichart, G. J., and Poulton, S. W.: Sedimentary phosphorus and iron cycling in and below the oxygen minimum zone of the northern Arabian Sea, Biogeosciences, 9, 2603-2624, doi: 10.5194/bg-9-2603-2012, 2012.

Lenstra, W., Klomp, R., Molema, F., Behrends, T., and Slomp, C.: A sequential extraction procedure for particulate manganese and its application to coastal marine sediments, Chemical Geology, 584, 120538, 2021.

Lyons, T. W., Werne, J. P., Hollander, D. J., and Murray, R. W.: Contrasting sulfur geochemistry and Fe/Al and Mo/Al ratios across the last oxic-to-anoxic transition in the Cariaco Basin, Venezuela, Chemical Geology, 195, 131-157, doi: 10.1016/s0009-2541(02)00392-3, 2003.

März, C., Poulton, S. W., Beckmann, B., Kuster, K., Wagner, T., and Kasten, S.: Redox sensitivity of P cycling during marine black shale formation: Dynamics of sulfidic and anoxic, non-sulfidic bottom waters, Geochimica Et Cosmochimica Acta, 72, 3703-3717, doi: 10.1016/j.gca.2008.04.025, 2008.

Paul, K. M., van Helmond, N. A. G. M., Slomp, C. P., Jokinen, S. A., Virtasalo, J. J., Filipsson, H. L., and Jilbert, T.: Sedimentary molybdenum and uranium: Improving proxies for deoxygenation in coastal depositional environments, Chemical Geology, 615, 121203, doi: 10.1016/j.chemgeo.2022.121203, 2023.

Poulton, S. W. and Canfield, D. E.: Development of a sequential extraction procedure for iron: implications for iron partitioning in continentally derived particulates, Chemical geology, 214, 209-221, 2005.

Yano, M., Yasukawa, K., Nakamura, K., Ikehara, M., and Kato, Y.: Geochemical Features of Redox-Sensitive Trace Metals in Sediments under Oxygen-Depleted Marine Environments, Minerals, 10, doi: 10.3390/min10111021, 2020.